# Accelerated Zeroth-order Method for Non-Smooth Stochastic Convex Optimization Problem with Infinite Variance

**Nikita Kornilov**
MIPT, SkolTech
kornilov.nm@phystech.edu

**Ohad Shamir**
Weizmann Institute of Science
ohad.shamir@weizmann.ac.il

**Aleksandr Lobanov**
MIPT, ISP RAS
lobanov.av@mipt.ru

**Darina Dvinskikh**
HSE University,
ISP RAS
dmdvinskikh@hse.ru

**Alexander Gasnikov**
MIPT,
ISP RAS, SkolTech
gasnikov@yandex.ru

**Innokentiy Shibaev**
MIPT,
IITP RAS
innokentiy.shibayev@phystech.edu

**Eduard Gorbunov**
MBZUAI
eduard.gorbunov@mbzuai.ac.ae

**Samuel Horváth**
MBZUAI
samuel.horvath@mbzuai.ac.ae

## Abstract

In this paper, we consider non-smooth stochastic convex optimization with two function evaluations per round under infinite noise variance. In the classical setting when noise has finite variance, an optimal algorithm, built upon the batched accelerated gradient method, was proposed in [17]. This optimality is defined in terms of iteration and oracle complexity, as well as the maximal admissible level of adversarial noise. However, the assumption of finite variance is burdensome and it might not hold in many practical scenarios. To address this, we demonstrate how to adapt a refined clipped version of the accelerated gradient (Stochastic Similar Triangles) method from [35] for a two-point zero-order oracle. This adaptation entails extending the batching technique to accommodate infinite variance — a non-trivial task that stands as a distinct contribution of this paper.

## 1 Introduction

In this paper, we consider stochastic non-smooth convex optimization problem

$$\min_{x \in \mathbb{R}^d} \left\{ f(x) \stackrel{\text{def}}{=} \mathbb{E}_{\xi \sim \mathcal{D}} \left[ f(x, \xi) \right] \right\}, \tag{1}$$

where $f(x, \xi)$ is $M_2(\xi)$-Lipschitz continuous in $x$ w.r.t. the Euclidean norm, and the expectation $\mathbb{E}_{\xi \sim \mathcal{D}} \left[ f(x, \xi) \right]$ is with respect to a random variable $\xi$ with unknown distribution $\mathcal{D}$. The optimization is performed only by accessing two function evaluations per round rather than sub-gradients, i.e., for any pair of points $x, y \in \mathbb{R}^d$, an oracle returns $f(x, \xi)$ and $f(y, \xi)$ with the same $\xi$. The primary motivation for this gradient-free oracle arises from different applications where calculating gradients is computationally infeasible or even impossible. For instance, in medicine, biology, and physics, the objective function can only be computed through numerical simulation or as the result of a real experiment, i.e., automatic differentiation cannot be employed to calculate function derivatives. Usually, a black-box function we are optimizing is affected by stochastic or computational noise. This noise can arise naturally from modeling randomness within a simulation or by computer discretization.

37th Conference on Neural Information Processing Systems (NeurIPS 2023).

In classical setting, this noise has light tails. However, usually in black-box optimization, we know nothing about the function, only its values at the requested points are available/computable, so light tails assumption may be violated. In this case, gradient-free algorithms may diverge. We aim to develop an algorithm that is robust even to heavy-tailed noise, i.e., noise with infinite variance. In theory, one can consider heavy-tailed noise to simulate situations where noticeable outliers may occur (even if the nature of these outliers is non-stochastic). Therefore we relax classical finite variance assumption and consider less burdensome assumption of finite $\alpha$-th moment, where $\alpha \in (1, 2]$.

In machine learning, interest in gradient-free methods is mainly driven by the bandit optimization problem [14, 2, 5], where a learner engages in a game with an adversary: the learner selects a point $x$, and the adversary chooses a point $\xi$. The learner's goal is to minimize the average regret based solely on observations of function values (losses) $f(x, \xi)$. As feedback, at each round, the learner receives losses at two points. This corresponds to a zero-order oracle in stochastic convex optimization with two function evaluations per round. The vast majority of researches assume sub-Gaussian distribution of rewards. However, in some practical cases (e.g., in finance [33]) reward distribution has heavy-tails or can be adversarial. For the heavy-tailed bandit optimization, we refer to [9].

Two-point gradient-free optimization for non-smooth (strongly) convex objective is a well-researched area. Numerous algorithms have been proposed which are optimal with respect to two criteria: oracle and iteration complexity. For a detailed overview, see the recent survey [15] and the references therein. Optimal algorithms, in terms of oracle call complexity, are presented in [10, 36, 3]. The distinction between the number of successive iterations (which cannot be executed in parallel) and the number of oracle calls was initiated with the lower bound obtained in [6]. It culminated with the optimal results from [17], which provides an algorithm which is optimal in both criteria. Specifically, the algorithm produces $\hat{x}$, an $\varepsilon$-solution of (1), such that we can guarantee $\mathbb{E}[f(\hat{x})] - \min_{x \in \mathbb{R}^d} f(x)$ after

$$\sim d^{\frac{1}{4}} \varepsilon^{-1} \quad \text{successive iterations,}$$
$$\sim d\varepsilon^{-2} \quad \text{oracle calls (number of } f(x, \xi) \text{ calculations).}$$

The convergence guarantee for this optimal algorithm from [17] was established in the classical setting of *light-tailed* noise, i.e., when noise has finite variance: $\mathbb{E}_\xi[M_2(\xi)^2] < \infty$. However, in many modern learning problems the variance might not be finite, leading the aforementioned algorithms to potentially diverge. Indeed, *heavy-tailed* noise is prevalent in contemporary applications of statistics and deep learning. For example, heavy-tailed behavior can be observed in training attention models [41] and convolutional neural networks [37, 20]. Consequently, our goal is to develop an optimal algorithm whose convergence is not hampered by this restrictive assumption. To the best of our knowledge, no existing literature on gradient-free optimization allows for $\mathbb{E}_\xi[M_2(\xi)^2]$ to be infinite. Furthermore, convergence results for all these gradient-free methods were provided in expectation, that is, without (non-trivial) high-probability bounds. Although the authors of [17] mentioned (without proof) that their results can be formulated in high probability using [19], this aspect notably affects the oracle calls complexity bound and complicates the analysis.

A common technique to relax finite variance assumption is gradient clipping [31]. Starting from the work of [26] (see also [8, 19]), there has been increased interest in algorithms employing gradient clipping to achieve high-probability convergence guarantees for stochastic optimization problems with *heavy-tailed* noise. Particularly, in just the last two years there have been proposed

- an optimal algorithm with a general proximal setup for non-smooth stochastic convex optimization problems with infinite variance [39] that converges in expectation (also referenced in [27]),
- an optimal adaptive algorithm with a general proximal setup for non-smooth online stochastic convex optimization problems with infinite variance [42] that converges with high probability,
- optimal algorithms using the Euclidean proximal setup for both smooth and non-smooth stochastic convex optimization problems and variational inequalities with infinite variance [35, 30, 29] that converge with high probability,
- an optimal variance-adaptive algorithm with the Euclidean proximal setup for non-smooth stochastic (strongly) convex optimization problems with infinite variance [24] that converges with high probability.

None of these papers discuss a gradient-free oracle. Moreover, they do not incorporate acceleration (given the non-smooth nature of the problems) with the exception of [35]. Acceleration is a crucial

component to achieving optimal bounds on the number of successive iterations. However, the approach in [35] presumes smoothness and does not utilize batching. To apply the convergence results from [35] to our problem, we need to adjust our problem formulation to be smooth. This is achieved by using the Smoothing technique [27, 36, 17]. In the work [22] authors proposed an algorithm based on Smoothing technique and non-accelerated Stochastic Mirror Descent with clipping. However, this work also does not support acceleration, minimization on the whole space and batching. Adapting the technique from [35] to incorporate batching necessitates a substantial generalization. We regard this aspect of our work as being of primary interest.

Heavy-tailed noise can also be handled without explicit gradient clipping, e.g., by using Stochastic Mirror Descent algorithm with a particular class of uniformly convex mirror maps [39]. However, the convergence guarantee for this algorithm is given in expectation. Moreover, applying batching and acceleration is a non-trivial task. Without this, we are not able to get the optimal method in terms of the number of iterations and not only in terms of oracle complexity. There are also some studies on the alternatives to gradient clipping [21] but the results for these alternatives are given in expectation and are weaker than the state-of-the-art results for the methods with clipping. This is another reason why we have chosen gradient clipping to handle the heavy-tailed noise.

## 1.1 Contributions

We generalize the optimal result from [17] to accommodate a weaker assumption that allows the noise to exhibit *heavy tails*. Instead of the classical assumption of finite variance, we require the boundedness of the $\alpha$-moment: there exists $\alpha \in (1, 2]$ such that $\mathbb{E}_\xi[M_2(\xi)^\alpha] < \infty$. Notably, when $\alpha < 2$, this assumption is less restrictive than the assumption of a finite variance and thus it has garnered considerable interest recently, see [41, 35, 29] and the references therein. Under this assumption we prove that for convex $f$, an $\varepsilon$-solution can be found with *high probability* after

$$\sim d^{\frac{1}{4}} \varepsilon^{-1} \quad \text{successive iterations,}$$

$$\sim \left(\sqrt{d}/\varepsilon\right)^{\frac{\alpha}{\alpha-1}} \quad \text{oracle calls,}$$

and for $\mu$-strongly convex $f$, the $\varepsilon$-solution can be found with *high probability* after

$$\sim d^{\frac{1}{4}} \left(\mu\varepsilon\right)^{-\frac{1}{2}} \quad \text{successive iterations,}$$

$$\sim \left(d/(\mu\varepsilon)\right)^{\frac{\alpha}{2(\alpha-1)}} \quad \text{oracle calls.}$$

In both instances, the number of oracle calls is optimal in terms of $\varepsilon$-dependency within the non-smooth setting [27, 39]. For first-order optimization under *heavy-tailed* noise, the optimal $\varepsilon$-dependency remains consistent, as shown in [35, Table 1].

In what follows, we highlight the following several important aspects of our results

- **High-probability guarantees.** We provide upper-bounds on the number of iterations/oracle calls needed to find a point $\hat{x}$ such that $f(\hat{x}) - \min_{x \in \mathbb{R}^d} f(x) \leq \varepsilon$ with probability at least $1 - \beta$. The derived bounds have a poly-logarithmic dependence on $1/\beta$. To the best of our knowledge, there are no analogous high-probability results, even for noise with bounded variance.

- **Generality of the setup.** Our results are derived under the assumption that gradient-free oracle returns values of stochastic realizations $f(x, \xi)$ subject to (potentially adversarial) bounded noise. We further provide upper bounds for the magnitude of this noise, contingent upon the target accuracy $\varepsilon$ and confidence level $\beta$. Notably, our assumptions about the objective and noise are confined to a compact subset of $\mathbb{R}^d$. This approach, which differs from standard ones in derivative-free optimization, allows us to encompass a wide range of problems. This approach differs from standard ones in derivative-free optimization

- **Batching without bounded variance.** To establish the aforementioned upper bounds, we obtain the following: given $X_1, \ldots, X_B$ as independent random vectors in $\mathbb{R}^d$ where $\mathbb{E}[X_i] = x \in \mathbb{R}^d$ and $\mathbb{E}\|X_i - x\|_2^\alpha \leq \sigma^\alpha$ for some $\sigma \geq 0$ and $\alpha \in (1, 2]$, then

$$\mathbb{E}\left[\left\|\frac{1}{B}\sum_{i=1}^{B} X_i - x\right\|_2^\alpha\right] \leq \frac{2\sigma^\alpha}{B^{\alpha-1}}. \tag{2}$$

When $\alpha = 2$, this result aligns with the conventional case of bounded variance (accounting for a numerical factor of 2). Unlike existing findings, such as [40, Lemma 7] where $\alpha < 2$, the relation (2) does not exhibit a dependency on the dimension $d$. Moreover, (2) offers a theoretical basis to highlight the benefits of mini-batching, applicable to methods highlighted in this paper as well as first-order methods presented in [35, 29].

- **Dependency on $d$.** As far as we are aware, an open question remains: is the bound $\left(\sqrt{d}/\varepsilon\right)^{\frac{\alpha}{\alpha-1}}$ optimal regarding its dependence on $d$? For smooth stochastic convex optimization problems using a $(d+1)$-points stochastic zeroth-order oracle, the answer is negative. The optimal bound is proportional to $d\varepsilon^{-\frac{\alpha}{\alpha-1}}$. Consequently, for $\alpha \in (1,2)$, our results are intriguing because the dependence on $d$ in our bound differs from known results in the classical case where $\alpha = 2$.

## 1.2 Paper organization

The paper is organized as follows. In Section 2, we give some preliminaries, such as smoothing technique and gradient estimation, that are workhorse of our algorithms. Section 3 is the main section presenting two novel gradient-free algorithms along with their convergence results in high probability. These algorithms solve non-smooth stochastic optimization under *heavy-tailed* noise, and they will be reffered to as ZO-clipped-SSTM and R-ZO-clipped-SSTM (see Algorithms 1 and 2 respectively). In Section 4, we extend our results to gradient-free oracle corrupted by additive deterministic adversarial noise. In Section 5, we describe the main ideas behind the proof and emphasize key lemmas. In Section 6, we provide numerical experiments on the synthetic task that demonstrate the robustness of the proposed algorithms towards heavy-tailed noise.

## 2 Preliminaries

**Assumptions on a subset.** Although we consider an unconstrained optimization problem, our analysis does not require any assumptions to hold on the entire space. For our purposes, it is sufficient to introduce all assumptions only on some convex set $Q \in \mathbb{R}^d$ since we prove that the considered methods do not leave some ball around the solution or some level-set of the objective function with high probability. This allows us to consider fairly large classes of problems.

**Assumption 1 (Strong convexity)** *There exist a convex set $Q \subset \mathbb{R}^d$ and $\mu \geq 0$ such that function $f(x, \xi)$ is $\mu$-strongly convex on $Q$ for any fixed $\xi$, i.e.*

$$f(\lambda x_1 + (1-\lambda)x_2) \leq \lambda f(x_1) + (1-\lambda)f(x_2) - \frac{1}{2}\mu\lambda(1-\lambda)\|x_1 - x_2\|_2^2,$$

*for all $x_1, x_2 \in Q, \lambda \in [0,1]$.*

This assumption implies that $f(x)$ is $\mu$-strongly convex as well.

For a small constant $\tau > 0$, let us define an expansion of set $Q$ namely $Q_\tau = Q + B_2^d$, where $+$ stands for Minkowski addition. Using this expansion we make the following assumption.

**Assumption 2 (Lipschitz continuity and boundedness of $\alpha$-moment)** *There exist a convex set $Q \subset \mathbb{R}^d$ and $\tau > 0$ such that function $f(x, \xi)$ is $M_2(\xi)$-Lipschitz continuous w.r.t. the Euclidean norm on $Q_\tau$, i.e., for all $x_1, x_2 \in Q_\tau$*

$$|f(x_1, \xi) - f(x_2, \xi)| \leq M_2(\xi)\|x_1 - x_2\|_2.$$

*Moreover, there exist $\alpha \in (1, 2]$ and $M_2 > 0$ such that $\mathbb{E}_\xi[M_2(\xi)^\alpha] \leq M_2^\alpha$.*

If $\alpha < 2$, we say that noise is *heavy-tailed*. When $\alpha = 2$, the above assumption recovers the standard uniformly bounded variance assumption.

**Lemma 1** *Assumption 2 implies that $f(x)$ is $M_2$-Lipschitz on $Q$.*

**Randomized smoothing.** The main scheme that allows us to develop batch-parallel gradient-free methods for non-smooth convex problems is randomized smoothing [13, 17, 27, 28, 38] of a non-smooth function $f(x, \xi)$. The smooth approximation to a non-smooth function $f(x, \xi)$ is defined as

$$\hat{f}_\tau(x) \stackrel{\text{def}}{=} \mathbb{E}_{\mathbf{u},\xi}[f(x + \tau\mathbf{u}, \xi)], \tag{3}$$

where $\mathbf{u} \sim U(B_2^d)$ is a random vector uniformly distributed on the Euclidean unit ball $B_2^d \stackrel{\text{def}}{=} \{x \in \mathbb{R}^d : \|x\|_2 \leq 1\}$. In this approximation, a new type of randomization appears in addition to stochastic variable $\xi$.

The next lemma gives estimates for the quality of this approximation. In contrast to $f(x)$, function $\hat{f}_\tau(x)$ is smooth and has several useful properties.

**Lemma 2** *[17, Theorem 2.1.] Let there exist a subset $Q \subset \mathbb{R}^d$ and $\tau > 0$ such that Assumptions 1 and 2 hold on $Q_\tau$. Then,*

1. *Function $\hat{f}_\tau(x)$ is convex, Lipschitz with constant $M_2$ on $Q$, and satisfies*

$$\sup_{x \in Q} |\hat{f}_\tau(x) - f(x)| \leq \tau M_2.$$

2. *Function $\hat{f}_\tau(x)$ is differentiable on $Q$ with the following gradient*

$$\nabla \hat{f}_\tau(x) = \mathbb{E}_{\mathbf{e}} \left[ \frac{d}{\tau} f(x + \tau\mathbf{e})\mathbf{e} \right],$$

   *where $\mathbf{e} \sim U(S_2^d)$ is a random vector uniformly distributed on unit Eucledian sphere $S_2^d \stackrel{\text{def}}{=} \{x \in \mathbb{R}^d : \|x\|_2 = 1\}$.*

3. *Function $\hat{f}_\tau(x)$ is L-smooth with $L = \sqrt{d}M_2/\tau$ on $Q$.*

Our algorithms will aim at minimizing the smooth approximation $\hat{f}_\tau(x)$. Given Lemma 2, the output of the algorithm will also be a good approximate minimizer of $f(x)$ when $\tau$ is sufficiently small.

**Gradient estimation.** Our algorithms will based on randomized gradient estimate, which will then be used in a first order algorithm. Following [36], the gradient can be estimated by the following vector:

$$g(x, \xi, \mathbf{e}) = \frac{d}{2\tau}(f(x + \tau\mathbf{e}, \xi) - f(x - \tau\mathbf{e}, \xi))\mathbf{e}, \tag{4}$$

where $\tau > 0$ and $\mathbf{e} \sim U(S_2^d)$. We also use batching technique in order to allow parallel calculation of gradient estimation and acceleration. Let $B$ be a batch size, we sample $\{\mathbf{e}_i\}_{i=1}^B$ and $\{\xi_i\}_{i=1}^B$ independently, then

$$g^B(x, \{\xi\}, \{\mathbf{e}\}) = \frac{d}{2B\tau} \sum_{i=1}^B (f(x + \tau\mathbf{e}_i, \xi_i) - f(x - \tau\mathbf{e}_i, \xi_i))\mathbf{e}_i. \tag{5}$$

The next lemma states that $g^B(x, \{\xi\}, \{\mathbf{e}\})$ from (5) is an unbiased estimate of the gradient of $\hat{f}_\tau(x)$ (3). Moreover, under heavy-tailed noise (Assumption 2) with bounded $\alpha$-moment $g^B(x, \{\xi\}, \{\mathbf{e}\})$ has also bounded $\alpha$-moment.

**Lemma 3** *Under Assumptions 1 and 2, it holds*

$$\mathbb{E}_{\xi,\mathbf{e}}[g(x, \xi, \mathbf{e})] = \mathbb{E}_{\{\xi\},\{\mathbf{e}\}}[g^B(x, \{\xi\}, \{\mathbf{e}\})] = \nabla \hat{f}_\tau(x).$$

*and*

$$\mathbb{E}_{\xi,\mathbf{e}}[\|g(x, \xi, \mathbf{e}) - \mathbb{E}_{\xi,\mathbf{e}}[g(x, \xi, \mathbf{e})]\|_2^\alpha] \leq \sigma^\alpha \stackrel{\text{def}}{=} \left( \frac{\sqrt{d}M_2}{2^{\frac{1}{4}}} \right)^\alpha,$$

$$\mathbb{E}_{\{\xi\},\{\mathbf{e}\}}[\|g^B(x, \{\xi\}, \{\mathbf{e}\}) - \mathbb{E}_{\{\xi\},\{\mathbf{e}\}}[g^B(x, \{\xi\}, \{\mathbf{e}\})]\|_2^\alpha] \leq \frac{2\sigma^\alpha}{B^{\alpha-1}}. \tag{6}$$

# 3 Main Results

In this section, we present two our new zero-order algorithms, which we will refer to as ZO-clipped-SSTM and R-ZO-clipped-SSTM, to solve problem (1) under *heavy-tailed* noise assumption. To deal with heavy-tailed noise, we use clipping technique which clips heavy tails. Let $\lambda > 0$ be clipping constant and $g \in \mathbb{R}^d$, then clipping operator $\texttt{clip}$ is defined as

$$\texttt{clip}(g, \lambda) = \begin{cases} \frac{g}{\|g\|_2} \min(\|g\|_2, \lambda), & g \neq 0, \\ 0, & g = 0. \end{cases} \tag{7}$$

We apply clipping operator for batched gradient estimate $g^B(x, \{\xi\}, \{\mathbf{e}\})$ from (5) and then feed it into first-order Clipped Stochastic Similar Triangles Method (clipped-SSTM) from [18]. We will refer to our zero-order versions of clipped-SSTM as ZO-clipped-SSTM and R-ZO-clipped-SSTM for the convex case and strongly convex case respectively.

## 3.1 Convex case

Let us suppose that Assumption 1 is satisfied with $\mu = 0$.

---

**Algorithm 1** ZO-clipped-SSTM $\left(x^0, K, B, a, \tau, \{\lambda_k\}_{k=0}^{K-1}\right)$

---

**Input:** starting point $x^0$, number of iterations $K$, batch size $B$, stepsize $a > 0$, smoothing parameter $\tau$, clipping levels $\{\lambda_k\}_{k=0}^{K-1}$.
1: Set $L = \sqrt{d}M_2/\tau$, $A_0 = \alpha_0 = 0$, $y^0 = z^0 = x^0$
2: **for** $k = 0, \ldots, K-1$ **do**
3:      Set $\alpha_{k+1} = {k+2}/{2aL}$, $A_{k+1} = A_k + \alpha_{k+1}$.
4:      $x^{k+1} = \frac{A_k y^k + \alpha_{k+1} z^k}{A_{k+1}}$.
5:      Sample $\{\xi_i^k\}_{i=1}^B \sim \mathcal{D}$ and $\{\mathbf{e}_i^k\}_{i=1}^B \sim S_2^d$ independently.
6:      Compute gradient estimation $g^B(x^{k+1}, \{\xi^k\}, \{\mathbf{e}^k\})$ as defined in (5).
7:      Compute clipped $\tilde{g}_{k+1} = \texttt{clip}\left(g^B(x^{k+1}, \{\xi^k\}, \{\mathbf{e}^k\}), \lambda_k\right)$ as defined in (7).
8:      $z^{k+1} = z^k - \alpha_{k+1}\tilde{g}_{k+1}$.
9:      $y^{k+1} = \frac{A_k y^k + \alpha_{k+1} z^{k+1}}{A_{k+1}}$.
10: **end for**
**Output:** $y^K$

---

**Theorem 1 (Convergence of ZO-clipped-SSTM)** *Let Assumptions 1 and 2 hold with $\mu = 0$ and $Q = \mathbb{R}^d$. Let $\|x^0 - x^*\|^2 \leq R^2$, where $x^0$ is a starting point and $x^*$ is an optimal solution to (1). Then for the output $y^K$ of* ZO-clipped-SSTM*, run with batchsize $B$, $A = \ln{4K}/\beta \geq 1$, $a = \Theta(\max\{A^2, \sqrt{d}M_2 K^{\frac{(\alpha+1)}{\alpha}} A^{\frac{(\alpha-1)}{\alpha}}/LRB^{\frac{(\alpha-1)}{\alpha}}\}), \tau = \varepsilon/4M_2$ and $\lambda_k = \Theta(R/(\alpha_{k+1}A))$, we can guarantee $f(y^K) - f(x^*) \leq \varepsilon$ with probability at least $1 - \beta$ ( for any $\beta \in (0, 1]$) after*

$$K = \widetilde{\mathcal{O}}\left(\max\left\{\frac{M_2\sqrt[4]{d}R}{\varepsilon}, \frac{1}{B}\left(\frac{\sqrt{d}M_2 R}{\varepsilon}\right)^{\frac{\alpha}{\alpha-1}}\right\}\right) \tag{8}$$

*successive iterations and $K \cdot B$ oracle calls. Moreover, with probability at least $1 - \beta$ the iterates of* ZO-clipped-SSTM *remain in the ball $B_{2R}(x^*)$, i.e., $\{x^k\}_{k=0}^{K+1}, \{y^k\}_{k=0}^K, \{z^k\}_{k=0}^K \subseteq B_{2R}(x^*)$.*

Here and below notation $\widetilde{\mathcal{O}}$ means an upper bound on the growth rate hiding logarithms. The first term in bound (8) is optimal for the deterministic case for non-smooth problem (see [6]) and the second term in bound (8) is optimal in $\varepsilon$ for $\alpha \in (1, 2]$ and zero-point oracle (see [27]).

We notice that increasing the batch size $B$ to reduce the number of successive iterations makes sense only as long as the first term of (8) lower than the second one, i.e. there exists optimal value of batchsize

$$B \leq \left(\frac{\sqrt{d}M_2 R}{\varepsilon}\right)^{\frac{1}{\alpha-1}}.$$

## 3.2 Strongly-convex case

Now we suppose that Assumption 1 is satisfied with $\mu > 0$. For this case we employ ZO-clipped-SSTM with restarts technique. We will call this algorithm as R-ZO-clipped-SSTM (see Algorithm 2). At each round R-ZO-clipped-SSTM call ZO-clipped-SSTM with starting point $\hat{x}^t$, which is the output from the previous round, and for $K_t$ iterations.

---

**Algorithm 2** R-ZO-clipped-SSTM

---

**Input:** starting point $x^0$, number of restarts $N$, number of steps $\{K_t\}_{t=1}^N$, batchsizes $\{B_t\}_{t=1}^N$, stepsizes $\{a_t\}_{t=1}^N$, smoothing parameters $\{\tau_t\}_{t=1}^N$, clipping levels $\{\lambda_k^1\}_{k=0}^{K_1-1}, ..., \{\lambda_k^N\}_{k=0}^{K_N-1}$

1: $\hat{x}^0 = x^0$.
2: **for** $t = 1, \ldots, N$ **do**
3:     $\hat{x}^t = $ ZO-clipped-SSTM $\left( \hat{x}^{t-1}, K_t, B_t, a_t, \tau_t, \{\lambda_k^t\}_{k=0}^{K_t-1} \right)$.
4: **end for**
**Output:** $\hat{x}^N$

---

**Theorem 2 (Convergence of R-ZO-clipped-SSTM)** *Let Assumptions 1, 2 hold with $\mu > 0$ and $Q = \mathbb{R}^d$. Let $\|x^0 - x^*\|^2 \le R^2$, where $x^0$ is a starting point and $x^*$ is the optimal solution to (1). Let also $N = \lceil \log_2(\mu R^2/2\varepsilon) \rceil$ be the number of restarts. Let at each stage $t = 1, ..., N$ of R-ZO-clipped-SSTM, ZO-clipped-SSTM is run with batchsize $B_t$, $\tau_t = \varepsilon_t/4M_2$, $L_t = M_2\sqrt{d}/\tau_t$, $K_t = \widetilde{\Theta}(\max\{\sqrt{L_t R_{t-1}^2/\varepsilon_t}, (\sigma R_{t-1}/\varepsilon_t)^{\frac{\alpha}{(\alpha-1)}}/B_t\})$, $a_t = \widetilde{\Theta}(\max\{1, \sigma K_t^{\frac{\alpha+1}{\alpha}}/L_t R_t\})$ and $\lambda_k^t = \widetilde{\Theta}(R/\alpha_{k+1}^t)$, where $R_{t-1} = 2^{-\frac{(t-1)}{2}}R$, $\varepsilon_t = \mu R_{t-1}^2/4$, $\ln^{4NK_t}/\beta \ge 1$, $\beta \in (0,1]$. Then to guarantee $f(\hat{x}^N) - f(x^*) \le \varepsilon$ with probability at least $1 - \beta$, R-ZO-clipped-SSTM requires*

$$\widetilde{\mathcal{O}} \left( \max \left\{ \sqrt{\frac{M_2^2\sqrt{d}}{\mu\varepsilon}}, \left( \frac{dM_2^2}{\mu\varepsilon} \right)^{\frac{\alpha}{2(\alpha-1)}} \right\} \right) \tag{9}$$

*oracle calls. Moreover, with probability at least $1 - \beta$ the iterates of R-ZO-clipped-SSTM at stage $t = 1, \ldots, N$ stay in the ball $B_{2R_{t-1}}(x^*)$.*

The obtained complexity bound (see the proof in Appendix C.2) is the first optimal (up to logarithms) high-probability complexity bound under Assumption 2 for the smooth strongly convex problems. Indeed, the first term cannot be improved in view of the deterministic lower bound [27], and the second term is optimal [41].

## 4 Setting with Adversarial Noise

Often, black-box access of $f(x, \xi)$ are affected by some deterministic noise $\delta(x)$. Thus, now we suppose that a zero-order oracle instead of objective values $f(x, \xi)$ returns its noisy approximation

$$f_\delta(x, \xi) \stackrel{\text{def}}{=} f(x, \xi) + \delta(x). \tag{10}$$

This noise $\delta(x)$ can be interpreted, e.g., as a computer discretization when calculating $f(x, \xi)$. For our analysis, we need this noise to be uniformly bounded. Recently, noisy «black-box» optimization with bounded noise has been actively studied [11, 25]. The authors of [11] consider deterministic adversarial noise, while in [25] stochastic adversarial noise was explored.

**Assumption 3 (Boundedness of noise)** *There exists a constant $\Delta > 0$ such that $|\delta(x)| \le \Delta$ for all $x \in Q$.*

This is a standard assumption often used in the literature (e.g., [17]). Moreover, in some applications [4] the bigger the noise the cheaper the zero-order oracle. Thus, it is important to understand the maximum allowable level of adversarial noise at which the convergence of the gradient-free algorithm is unaffected.

## 4.1 Non-smooth setting

In noisy setup, gradient estimate from (4) is replaced by

$$g(x, \xi, \mathbf{e}) = \frac{d}{2\tau} \left( f_\delta(x + \tau\mathbf{e}, \xi) - f_\delta(x - \tau\mathbf{e}, \xi) \right) \mathbf{e}. \tag{11}$$

Then (6) from Lemma 3 will have an extra factor driven by noise (see Lemma 2.3 from [22])

$$\mathbb{E}_{\{\xi\},\{\mathbf{e}\}} \left[ \left\| g^B(x, \{\xi\}, \{\mathbf{e}\}) - \mathbb{E}_{\{\xi\},\{\mathbf{e}\}}[g^B(x, \{\xi\}, \{\mathbf{e}\})] \right\|_2^\alpha \right] \le \frac{2}{B^{\alpha-1}} \left( \frac{\sqrt{d}M_2}{2^{\frac{1}{4}}} + \frac{d\Delta}{\tau} \right)^\alpha.$$

To guarantee the same convergence of the algorithm as in Theorem 1 (see (8)) for the adversarial deterministic noise case, the variance term must dominate the noise term, i.e. $d\Delta\tau^{-1} \lesssim \sqrt{d}M_2$. Note that if the term with noise dominates the term with variance, it does not mean that the gradient-free algorithm will not converge. In contrast, algorithm will still converge, only slower (oracle complexity will be $\sim \varepsilon^{-2}$ times higher). Thus, if we were considering the zero-order oracle concept with adversarial stochastic noise, it would be enough to express the noise level $\Delta$, and substitute the value of the smoothing parameter $\tau$ to obtain the maximum allowable noise level. However, since we are considering the concept of adversarial noise in a deterministic setting, following previous work [11, 1] we can say that adversarial noise accumulates not only in the variance, but also in the bias:

$$\mathbb{E}_{\xi,\mathbf{e}}\langle [g(x, \xi, \mathbf{e})] - \nabla\hat{f}_\tau(x), r \rangle \lesssim \sqrt{d}\Delta\|r\|_2\tau^{-1}, \qquad \text{for all } r \in \mathbb{R}^d.$$

This bias can be controlled by the noise level $\Delta$, i.e., in order to achieve the $\varepsilon$-accuracy algorithm considered in this paper, the noise condition must be satisfied:

$$\Delta \lesssim \frac{\tau\varepsilon}{R\sqrt{d}}. \tag{12}$$

As we can see, we have a more restrictive condition on the noise level in the bias (12) than in the variance ($\Delta \lesssim \gamma M_2/\sqrt{d}$). Thus, the maximum allowable level of adversarial deterministic noise, which guarantees the same convergence of ZO-clipped-SSTM, as in Theorem 1 (see (8)) is as follows

$$\Delta \lesssim \frac{\varepsilon^2}{RM_2\sqrt{d}},$$

where $\tau = \varepsilon/2M_2$ the smoothing parameter from Lemma 2.

**Remark 1 ($\mu$-strongly convex case)** *Let us assume that $f(x)$ is also $\mu$-strongly convex (see Assumption 1). Then, following the works [11, 22], we can conclude that the* R-ZO-clipped-SSTM *has the same oracle and iteration complexity as in Theorem 2 at the following maximum allowable level of adversarial noise: $\Delta \lesssim \mu^{1/2}\varepsilon^{3/2}/\sqrt{d}M_2$.*

## 4.2 Smooth setting

Now we examine the maximum allowable level of noise at which we can solve optimization problem (1) with $\varepsilon$-precision under the following additional assumption

**Assumption 4 (Smoothness)** *The function $f$ is $L$-smooth, i.e., it is differentiable on $Q$ and for all $x, y \in Q$ with $L > 0$:*
$$\|\nabla f(y) - \nabla f(x)\|_2 \le L\|y - x\|_2.$$

If Assumption 4 holds, then Lemma 2 can be rewritten as

$$\sup_{x \in Q} |\hat{f}_\tau(x) - f(x)| \le \frac{L\tau^2}{2}.$$

Thus, we can now present the convergence results of the gradient-free algorithm in the smooth setting. Specifically, if the Assumptions 2-4 are satisfied, then ZO-clipped-SSTM converges to $\varepsilon$-accuracy after $K = \widetilde{\mathcal{O}}\left(\sqrt{LR^2\varepsilon^{-1}}\right)$ iterations with probability at least $1 - \beta$. It is easy to see that the iteration complexity improves in the smooth setting (since the Lipschitz gradient constant $L$ already exists,

i.e., no smoothing is needed), but oracle complexity remained unchanged (since we are still using the gradient approximation via $l_2$ randomization (11) instead of the true gradient $\nabla f(x)$), consistent with the already optimal estimate on oracle calls: $\widetilde{\mathcal{O}}\left(\left(\sqrt{d}M_2R\varepsilon^{-1}\right)^{\frac{\alpha}{\alpha-1}}\right)$. And to obtain the maximum allowable level of adversarial noise $\Delta$ in the smooth setting, which guarantees such convergence, it is sufficient to substitute the smoothing parameter $\tau = \sqrt{\varepsilon/L}$ in the inequality (12):

$$\Delta \lesssim \frac{\varepsilon^{3/2}}{R\sqrt{dL}}.$$

Thus, we can conclude that smooth setting improves iteration complexity and the maximum allowable noise level for the gradient-free algorithm, but the oracle complexity remains unchanged.

**Remark 2 ($\mu$-strongly convex case)** *Suppose that $f(x)$ is also $\mu$-strongly convex (see Assumption 1). Then we can conclude that R-ZO-clipped-SSTM has the oracle and iteration complexity just mentioned above at the following maximum allowable level of adversarial noise: $\Delta \lesssim \mu^{1/2}\varepsilon/\sqrt{dL}$.*

**Remark 3 (Upper bounds optimality)** *The upper bounds on maximum allowable level of adversarial noise obtained in this section in both non-smooth and smooth settings are optimal in terms of dependencies on $\varepsilon$ and $d$ according to the works [32, 34].*

**Remark 4 (Better oracle complexity)** *In the aforementioned approach in the case when $f(x,\xi)$ has Lipschitz gradient in $x$ (for all $\xi$) one can improve oracle complexity from $\widetilde{\mathcal{O}}\left(\left(\sqrt{d}M_2R\varepsilon^{-1}\right)^{\frac{\alpha}{\alpha-1}}\right)$ to $\widetilde{\mathcal{O}}\left(d\left(M_2R\varepsilon^{-1}\right)^{\frac{\alpha}{\alpha-1}}\right)$. This can be done by using component-wise finite-difference stochastic gradient approximation [15]. Iteration complexity remains $\widetilde{\mathcal{O}}\left(\sqrt{LR^2\varepsilon^{-1}}\right)$. The same can be done for $\mu$-strongly convex case: from $\widetilde{\mathcal{O}}\left(\left(dM_2^2(\mu\varepsilon)^{-1}\right)^{\frac{\alpha}{2(\alpha-1)}}\right)$ to $\widetilde{\mathcal{O}}\left(d\left(M_2^2(\mu\varepsilon)^{-1}\right)^{\frac{\alpha}{2(\alpha-1)}}\right)$.*

# 5 Details of the proof

The proof is built upon a combination of two techniques. The first one is the Smoothing technique from [17] that is used to develop a gradient-free method for convex non-smooth problems based on full-gradient methods. The second technique is the Accelerated Clipping technique that has been recently developed for smooth problems with the noise having infinite variance and first-order oracle [35]. The authors of [35] propose clipped-SSTM method that we develop in our paper. We modify clipped-SSTM by introducing batching into it. Note that due to the infinite variance, such a modification is interesting in itself. Then we run batched clipped-SSTM with gradient estimations of function $f$ obtained via Smoothing technique and two-point zeroth-order oracle. To do this, we need to estimate the variance of the clipped version of the batched gradient estimation.

In more detail, we replace the initial problem of minimizing $f$ by minimizing its smoothed approximation $\hat{f}_\tau$, see Lemma 2 In order to use estimated gradient of $\hat{f}_\tau$ defined in (4) or (5), we make sure that it has bounded $\alpha$-th moment. For these purposes we proof Lemma 3. First part shows boundness of unbatched estimated gradient $g$ defined in (4). It follows from measure concentration phenomenon for the Euclidean sphere for $\frac{M_2}{\tau}$-Lipschitz function $f(x + e\tau)$. According to this phenomenon probability of the functions deviation from its math expectation becomes exponentially small and $\alpha$-th moment of this deviation becomes bounded. Furthermore, the second part of Lemma 3 shows that batching helps to bound $\alpha$-th moment of batched gradient $g^B$ defined in (5) even more. Also batching allows parallel calculations reducing number of necessary iteration with the same number of oracle calls. All this is possible thanks to the result, interesting in itself, presented in the following Lemma.

**Lemma 4** *Let $X_1, \ldots, X_B$ be $d$-dimensional martingale difference sequence (i.e. $\mathbb{E}[X_i|X_{i-1}, \ldots, X_1] = 0$ for $1 < i \leq B$) satisfying for $1 \leq \alpha \leq 2$*
$$\mathbb{E}[\|X_i\|^\alpha|X_{i-1}, \ldots, X_1] \leq \sigma^\alpha.$$
*Then we have*
$$\mathbb{E}\left[\left\|\frac{1}{B}\sum_{i=1}^{B}X_i\right\|_2^\alpha\right] \leq \frac{2\sigma^\alpha}{B^{\alpha-1}}.$$

Next, we use the clipped-SSTM for function $\hat{f}_\tau$ with heavy-tailed gradient estimates. This algorithm was initially proposed for smooth functions in the work [35]. The scheme for proving convergence with high probability is also taken from it, the only difference is additional randomization caused by smoothing scheme.

## 6  Numerical experiments

We tested ZO-clipped-SSTM on the following problem
$$\min_{x\in\mathbb{R}^d} \|Ax - b\|_2 + \langle \xi, x \rangle,$$
where $\xi$ is a random vector with independent components sampled from the symmetric Levy $\alpha$-stable distribution with $\alpha = 3/2$, $A \in \mathbb{R}^{m\times d}$, $b \in \mathbb{R}^m$ (we used $d = 16$ and $m = 500$). For this problem, Assumption 1 holds with $\mu = 0$ and Assumption 2 holds with $\alpha = 3/2$ and $M_2(\xi) = \|A\|_2 + \|\xi\|_2$.

We compared ZO-clipped-SSTM, proposed in this paper, with ZO-SGD and ZO-SSTM. For these algorithms, we gridsearched batchsize $B$ : $5, 10, 50, 100, 500$ and stepsize $\gamma : 1e-3, 1e-4, 1e-5, 1e-6$. The best convergence was achieved with the following parameters:

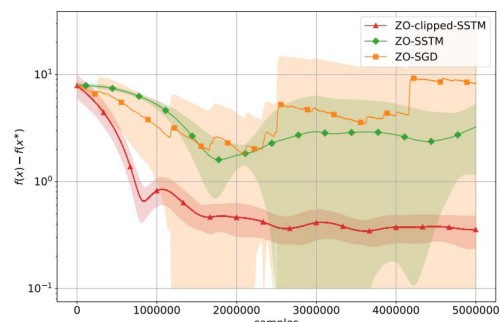

- ZO-clipped-SSTM: $\gamma = 1e - 3$, $B = 10$, $\lambda = 0.01$,
- ZO-SSTM: $\gamma = 1e - 5$, $B = 500$,
- ZO-SGD: $\gamma = 1e - 4$, $B = 100$, $\omega = 0.9$, where $\omega$ is a heavy-ball momentum parameter.

Figure 1: Convergence of ZO-clipped-SSTM, ZO-SGD and ZO-clipped-SSTM in terms of a gap function w.r.t. the number of consumed samples.

The code is written on Pythone and is publicly available at https://github.com/ClippedStochasticMethods/ZO-clipped-SSTM. Figure 1 presents the comparison of convergences averaged over 15 launches with different noise. In contrast to ZO-clipped-SSTM, the last two methods are unclipped and therefore failed to converge under haivy-tailed noise.

## 7  Conclusion and future directions

In this paper, we propose a first gradient-free algorithm ZO-clipped-SSTM and to solve problem (1) under *heavy-tailed* noise assumption. By using the restart technique we extend this algorithm for strongly convex objective, we refer to this algorithm as R-ZO-clipped-SSTM. The proposed algorithms are optimal with respect to oracle complexity (in terms of the dependence on $\varepsilon$), iteration complexity and the maximal level of noise (possibly adversarial). The algorithms can be adapted to composite and distributed minimization problems, saddle-point problems, and variational inequalities. Despite the fact that the algorithms utilize the two-point feedback, they can be modified to the one-point feedback. We leave it for future work.

Moreover, we provide theoretical basis to demonstrate benefits of batching technique in case of heavy-tailed stochastic noise and apply it to methods from this paper. Thanks to this basis, it is possible to use batching in other methods with heavy-tiled noise, e.g. first-order methods presented in [35, 29].

## 8  Acknowledgment

The work of Alexander Gasnikov, Aleksandr Lobanov, and Darina Dvinskikh was supported by a grant for research centers in the field of artificial intelligence, provided by the Analytical Center for the Government of the Russian Federation in accordance with the subsidy agreement (agreement identifier 000000D730321P5Q0002) and the agreement with the Ivannikov Institute for System Programming of dated November 2, 2021 No. 70-2021-00142.

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

## A Batching with unbounded variance

To prove batching Lemma 4 we generalize Lemma 4.2 from [7] not only for i.i.d. random variables with zero mean but for martingale difference sequence.

**Lemma 5** *Let $g(x) = sign(x)|x|^{\alpha-1} = \nabla\left(\frac{|x|^\alpha}{\alpha}\right)$ for $1 < \alpha \leq 2$. Then for any $h \geq 0$*

$$\max_h g(x+h) - g(x) = 2^{2-\alpha}h^{\alpha-1} = 2^{2-\alpha}g(h).$$

*Proof.* Consider $l(x) = g(x+h) - g(x)$. We see that $l$ is differentiable everywhere except $x = 0$ and $x = -h$. As long as $x \neq 0, -h$, we have

$$l'(x) = g'(x+h) - g'(x) = (\alpha-1)(|x+h|^{\alpha-2} - |x|^{\alpha-2}).$$

Since, we have $\alpha > 1$, $x = -\frac{h}{2}$ is local maxima for $l(x)$. Furthermore, note that $l'(x) \geq 0$ for $x \in \left(-\infty, -\frac{h}{2}\right) \setminus \{-h\}$ and $l'(x) \leq 0$ for $x \in \left(-\frac{h}{2}, \infty\right) \setminus \{0\}$. Therefore, $\frac{-h}{2}$ is global maxima.

**Lemma 6** *Let $x_1, \ldots, x_B$ be one-dimensional martingale difference sequence, i.e. $\mathbb{E}[x_i|x_{i-1}, \ldots, x_1] = 0$ for $1 < i \leq B$, satisfying for $1 \leq \alpha \leq 2$*

$$\mathbb{E}[|x_i|^\alpha|x_{i-1}, \ldots, x_1] \leq \sigma^\alpha.$$

*We have:*

$$\mathbb{E}\left[\left|\frac{1}{B}\sum_{i=1}^{B}x_i\right|^\alpha\right] \leq \frac{2\sigma^\alpha}{B^{\alpha-1}}.$$

*Proof.* Everywhere below we will use the following notations.

$$\mathbb{E}_{<i}[\cdot] \stackrel{\text{def}}{=} \mathbb{E}_{x_{i-1},\ldots,x_1}[\cdot], \qquad \mathbb{E}_{|<i}[\cdot] \stackrel{\text{def}}{=} \mathbb{E}[\cdot|x_{i-1},\ldots,x_1].$$

For $\alpha = 1$ proof follows from triangle inequality for $|\cdot|$. When $\alpha > 1$, we start by defining

$$S_i = \sum_{j=1}^{i} x_j, \qquad S_0 = 0, \qquad f(x) = |x|^\alpha.$$

Then we can calculate desired expectation as

$$
\begin{aligned}
\mathbb{E}[f(S_B)] &= \mathbb{E}\left[\sum_{i=1}^{B} f(S_i) - f(S_{i-1})\right] = \sum_{i=1}^{B} \mathbb{E}\left[f(S_i) - f(S_{i-1})\right] \\
&= \sum_{i=1}^{B} \mathbb{E}\left[\int_{S_{i-1}}^{S_i} f'(x)dx\right] \\
&= \sum_{i=1}^{B} \mathbb{E}\left[x_i f'(S_{i-1}) + \int_{S_{i-1}}^{S_i} f'(x) - f'(S_{i-1})dx\right].
\end{aligned}
\tag{13}
$$

While $\{x_i\}$ is martingale difference sequence then $\mathbb{E}[x_i f'(S_{i-1})] = \mathbb{E}_{<i}[\mathbb{E}_{|<i}[x_i f'(S_{i-1})]] = 0$. From (13) and Lemma 5 ($g(x) = f'(x)/\alpha$) we obtain

$$
\begin{aligned}
\mathbb{E}[f(S_B)] &= \sum_{i=1}^{B} \mathbb{E}\left[x_i f'(S_{i-1}) + \int_{S_{i-1}}^{S_i} f'(x) - f'(S_{i-1})dx\right] \leq 2^{1-\alpha}\sum_{i=1}^{B} \mathbb{E}\left[\int_{0}^{|x_i|} f'(t/2)dt\right] \\
&= \sum_{i=1}^{B} \mathbb{E}\left[\int_{0}^{|x_i|/2} 2f'(s)ds\right] = 2^{2-\alpha}\sum_{i=1}^{B} \mathbb{E}\left[f(|x_i|/2)\right] \\
&= 2^{2-\alpha}\sum_{i=1}^{B} \mathbb{E}_{<i}\left[\mathbb{E}_{|<i}[f(|x_i|/2)]\right] \leq 2B\sigma^\alpha.
\end{aligned}
\tag{14}
$$

Now we are ready to prove batching lemma for random variables with infinite variance.

**Lemma 7** *Let* $X_1, \ldots, X_B$ *be* $d$*-dimensional martingale difference sequence, i.e.,* $\mathbb{E}[X_i | X_{i-1}, \ldots, X_1] = 0$ *for* $1 < i \leq B$*, satisfying for* $1 \leq \alpha \leq 2$

$$\mathbb{E}[\|X_i\|_2^\alpha | X_{i-1}, \ldots, X_1] \leq \sigma^\alpha.$$

*We have*

$$\mathbb{E}\left[\left\|\frac{1}{B}\sum_{i=1}^B X_i\right\|_2^\alpha\right] \leq \frac{2\sigma^\alpha}{B^{\alpha-1}}.$$

*Proof.*

Let $g \sim \mathcal{N}(0, I)$ and $y_i \stackrel{def}{=} X_i^\top g$. Firstly, we prove that $\mathbb{E}[y_i^\alpha] \leq \mathbb{E}[\|X_i\|^\alpha]$. Indeed, using conditional expectation we get

$$
\begin{aligned}
\mathbb{E}[|y_i|^\alpha] &= \mathbb{E}\left[(|X_i^\top g|)^\alpha\right] = \mathbb{E}_{X_i}\left[\mathbb{E}_{g|X_i}\left[(|X_i^\top g|)^\alpha\right]\right] \\
&= \mathbb{E}_{X_i}\left[\mathbb{E}_{g|X_i}\left[((|X_i^\top g|)^2)^{\alpha/2}\right]\right] \\
&\stackrel{\text{Jensen inq}}{\leq} \mathbb{E}_{X_i}\left[\left(\mathbb{E}_{g|X_i}\left[(X_i^\top g)^2\right]\right)^{\alpha/2}\right] \\
&= \mathbb{E}_{X_i}\left[\left(\|X_i\|_2^2\right)^{\alpha/2}\right] = \mathbb{E}[\|X_i\|_2^\alpha].
\end{aligned}
\tag{15}
$$

Next, considering $X_i^\top g \sim \mathcal{N}(0, \|X\|^2)$ and, thus, $\mathbb{E}_g |X_i^\top g| = \|X\|$, we bound desired expectation as

$$\mathbb{E}_X\left[\left\|\sum_{i=1}^B X_i\right\|_2^\alpha\right] = \mathbb{E}_X\left[\left(\mathbb{E}_g\left|\sum_{i=1}^B X_i^\top g\right|\right)^\alpha\right] \tag{16}$$

$$\stackrel{\text{Jensen's inq}}{\leq} \mathbb{E}_{X,g}\left[\left|\sum_{i=1}^B X_i^\top g\right|^\alpha\right] = \mathbb{E}_{X,g}\left[\left|\sum_{i=1}^B y_i\right|^\alpha\right]. \tag{17}$$

Finally, we apply Lemma 6 for $y_i$ sequence with bounded $\alpha$-th moment from (15) and get

$$\mathbb{E}_X\left[\left\|\sum_{i=1}^B X_i\right\|_2^\alpha\right] \leq \mathbb{E}_{X,g}\left[\left|\sum_{i=1}^B y_i\right|^\alpha\right] \leq 2\sigma^\alpha B.$$

## B   Smoothing scheme

**Lemma 8** *Assumption 2 implies that* $f(x)$ *is* $M_2$ *Lipschitz on* $Q$*.*

*Proof.* For all $x_1, x_2 \in Q$

$$
\begin{aligned}
|f(x_1) - f(x_2)| &\leq |\mathbb{E}_\xi[f(x_1, \xi) - f(x_2, \xi)]| \leq \mathbb{E}_\xi[|f(x_1, \xi) - f(x_2, \xi)|] \tag{18} \\
&\leq \mathbb{E}_\xi[M_2(\xi)]\|x_1 - x_2\|_2 \leq M_2\|x_1 - x_2\|_2. \tag{19}
\end{aligned}
$$

The following lemma gives some facts about the measure concentration on the Euclidean unit sphere for next proof.

**Lemma 9** *Let* $f(x)$ *be* $M_2$ *Lipschitz continuous function w.r.t* $\|\cdot\|$*. If* $\mathbf{e}$ *is random and uniformly distributed on the Euclidean sphere and* $\alpha \in (1, 2]$*, then*

$$\mathbb{E}_\mathbf{e}\left[(f(\mathbf{e}) - \mathbb{E}_\mathbf{e}[f(\mathbf{e})])^{2\alpha}\right] \leq \left(\frac{bM_2^2}{d}\right)^\alpha, \quad b = \frac{1}{\sqrt{2}}.$$

*Proof.* A standard result of the measure concentration on the Euclidean unit sphere implies that $\forall t > 0$

$$Pr\left(|f(\mathbf{e}) - \mathbb{E}[f(\mathbf{e})]| > t\right) \leq 2\exp(-b'dt^2/M_2^2), \quad b' = 2 \tag{20}$$

(see the proof of Proposition 2.10 and Corollary 2.6 in [23]). Therefore,

$$
\begin{aligned}
\mathbb{E}_{\mathbf{e}}\left[(f(\mathbf{e}) - \mathbb{E}_{\mathbf{e}}[f(\mathbf{e})])^{2\alpha}\right] &= \int_{t=0}^{\infty} Pr\left(|f(\mathbf{e}) - \mathbb{E}[f(\mathbf{e})]|^{2\alpha} > t\right) dt \\
&= \int_{t=0}^{\infty} Pr\left(|f(\mathbf{e}) - \mathbb{E}[f(\mathbf{e})]| > t^{1/2\alpha}\right) dt \\
&\leq \int_{t=0}^{\infty} 2\exp\left(-b'd \cdot t^{1/\alpha}/M_2^2\right) dt \leq \left(\frac{bM_2^2}{d}\right)^{\alpha}. \qquad (21)
\end{aligned}
$$

Finally, below we prove Lemma 3 which states that batched gradient estimation from (5) has bounded $\alpha$-th moment.

*Proof of Lemma 3.*

1. We will prove equality immediately for $g^B$. Firstly, we notice that distribution of $\mathbf{e}$ is symmetric and by definition (5) we get

$$
\begin{aligned}
\mathbb{E}_{\xi,\mathbf{e}}[g^B(x,\xi,\mathbf{e})] &= \left(\frac{d}{2B\tau}\right) \sum_{i=1}^{B} \mathbb{E}_{\xi_i,\mathbf{e}_i}\left[f(x+\tau\mathbf{e}_i,\xi_i)\mathbf{e}_i - f(x-\tau\mathbf{e}_i,\xi_i)\mathbf{e_i}\right] \\
&= \frac{d}{B\tau} \sum_{i=1}^{B} \mathbb{E}_{\mathbf{e}_i}[\mathbb{E}_{\xi_i}[f(x+\tau\mathbf{e}_i,\xi_i)]\mathbf{e}_i] \\
&= \frac{d}{B\tau} \sum_{i=1}^{B} \mathbb{E}_{\mathbf{e}_i}[f(x+\tau\mathbf{e}_i)\mathbf{e}_i]. \qquad (22)
\end{aligned}
$$

   Using $\nabla\hat{f}_\tau(x) = \frac{d}{\tau}\mathbb{E}_{\mathbf{e}}[f(x+\tau\mathbf{e})\mathbf{e}]$ from Lemma 2 we obtain necessary result.

2. By definition (4) of estimated gradient $g$ we bound $\alpha$-th moment as

$$
\mathbb{E}_{\xi,\mathbf{e}}[\|g(x,\xi,\mathbf{e})\|^\alpha] = \mathbb{E}_{\xi,\mathbf{e}}\left[\left\|\frac{d}{2\tau}(f(x+\tau\mathbf{e},\xi) - f(x-\tau\mathbf{e},\xi))\mathbf{e}\right\|_2^\alpha\right]
$$

$$
= \left(\frac{d}{2\tau}\right)^\alpha \mathbb{E}_{\xi,\mathbf{e}}\left[\|\mathbf{e}\|_2^\alpha |f(x+\tau\mathbf{e},\xi) - f(x-\tau\mathbf{e},\xi)|^\alpha\right]. \qquad (23)
$$

   Considering $\|\mathbf{e}\|_2 = 1$ we can omit it. Next we add $\pm\delta(\xi)$ in (23) for all $\delta(\xi)$ and get

$$
\begin{aligned}
&\mathbb{E}_{\xi,\mathbf{e}}\left[|f(x+\tau\mathbf{e},\xi) - f(x-\tau\mathbf{e},\xi)|^\alpha\right] \\
&= \mathbb{E}_{\xi,\mathbf{e}}\left[|(f(x+\tau\mathbf{e},\xi) - \delta) - (f(x-\tau\mathbf{e},\xi) - \delta)|^\alpha\right].
\end{aligned}
$$

   Using Jensen's inequality for $|\cdot|^\alpha$ we bound

$$
\begin{aligned}
&\mathbb{E}_{\xi,\mathbf{e}}\left[|(f(x+\tau\mathbf{e},\xi) - \delta) - (f(x-\tau\mathbf{e},\xi) - \delta)|^\alpha\right] \\
&\leq 2^{\alpha-1}\mathbb{E}_{\xi,\mathbf{e}}\left[|f(x+\tau\mathbf{e},\xi) - \delta|^\alpha\right] + 2^{\alpha-1}\mathbb{E}_{\xi,\mathbf{e}}\left[|f(x-\tau\mathbf{e},\xi) - \delta|^\alpha\right].
\end{aligned}
$$

   We note that distribution of $\mathbf{e}$ is symmetric and add two terms together

$$
\begin{aligned}
&2^{\alpha-1}\mathbb{E}_{\xi,\mathbf{e}}\left[|f(x+\tau\mathbf{e},\xi) - \delta|^\alpha\right] + 2^{\alpha-1}\mathbb{E}_{\xi,\mathbf{e}}\left[|f(x-\tau\mathbf{e},\xi) - \delta|^\alpha\right] \\
&\leq 2^\alpha\mathbb{E}_{\xi,\mathbf{e}}\left[|f(x+\tau\mathbf{e},\xi) - \delta|^\alpha\right].
\end{aligned}
$$

   Let $\delta(\xi) = \mathbb{E}_{\mathbf{e}}[f(x+\tau\mathbf{e},\xi)]$, then because of Cauchy-Schwartz inequality and conditional expectation properties we obtain

$$
\begin{aligned}
2^\alpha\mathbb{E}_{\xi,\mathbf{e}}\left[|f(x+\tau\mathbf{e},\xi) - \delta|^\alpha\right] &= 2^\alpha\mathbb{E}_{\xi}\left[\mathbb{E}_{\mathbf{e}}\left[|f(x+\tau\mathbf{e},\xi) - \delta|^\alpha\right]\right] \\
&\leq 2^\alpha\mathbb{E}_{\xi}\left[\sqrt{\mathbb{E}_{\mathbf{e}}\left[|f(x+\tau\mathbf{e},\xi) - \mathbb{E}_{\mathbf{e}}[f(x+\tau\mathbf{e},\xi)]|^{2\alpha}\right]}\right].
\end{aligned}
$$

Next, we use Lemma 9 for $f(x + \tau\mathbf{e}, \xi)$ with fixed $\xi$ and Lipschitz constant $M_2(\xi)\tau$

$$2^\alpha \mathbb{E}_\xi \left[ \sqrt{\mathbb{E}_\mathbf{e} \left[ |f(x + \tau\mathbf{e}, \xi) - \mathbb{E}_\mathbf{e}[f(x + \tau\mathbf{e}, \xi)]|^{2\alpha} \right]} \right] \leq 2^\alpha \mathbb{E}_\xi \left[ \sqrt{\left( \frac{2^{-1/2}\tau^2 M_2^2(\xi)}{d} \right)^\alpha} \right]$$

$$= 2^\alpha \left( \frac{\tau^2 2^{-1/2}}{d} \right)^{\alpha/2} \mathbb{E}_\xi \left[ M_2^\alpha(\xi) \right] \leq 2^\alpha \left( \sqrt{\frac{2^{-1/2}}{d}} M_2\tau \right)^\alpha.$$

Finally, we get desired bound of estimated gradient

$$\mathbb{E}_{\xi,\mathbf{e}}[\|g(x, \xi, \mathbf{e})\|_2^\alpha] = \left( \frac{\sqrt{d}}{2^{1/4}} M_2 \right)^\alpha. \tag{24}$$

Now we apply Jensen inequality to

$$\mathbb{E}_{\xi,\mathbf{e}}[\|g(x, \xi, \mathbf{e}) - \mathbb{E}_{\xi,\mathbf{e}}[g(x, \xi, \mathbf{e})]\|_2^\alpha] \leq 2^{\alpha-1} \left( \mathbb{E}_{\xi,\mathbf{e}}[\|g(x, \xi, \mathbf{e})\|_2] + \mathbb{E}_{\xi,\mathbf{e}}[\|g(x, \xi, \mathbf{e})\|_2] \right) \tag{25}$$

And get necessary result.

For batched gradient $g^B$ we use Batching Lemma 7 and estimation (25).

## C   Missing Proofs for ZO-clipped-SSTM and R-ZO-clipped-SSTM

In this section, we provide the complete formulation of the main results for clipped-SSTM and R-clipped-SSTM and the missing proofs.

**Minimization on the subset** $Q$   In order to work with subsets of $Q \subseteq \mathbb{R}^d$ we must assume one more condition on $\hat{f}_\tau(x)$.

**Assumption 5** *We assume that there exist some convex set $Q \subseteq \mathbb{R}^d$, constants $\tau, L > 0$ such that for all $x, y \in Q$*

$$\|\nabla \hat{f}_\tau(x)\|_2^2 \quad \leq \quad 2L \left( \hat{f}_\tau(x) - \hat{f}_\tau^* \right), \tag{26}$$

*where $\hat{f}_\tau^* = \inf_{x \in Q} \hat{f}_\tau(x) > -\infty$.*

When $Q = \mathbb{R}^d$ (26) follows from Lemma 2 as well. But in general case this is not true. In work [35] in Section "Useful facts" authors show that, in the worst case, to have (26) on a set $Q$ one needs to assume smoothness on a slightly larger set.

Thus, in the full version of the theorems in which we can require much smaller $Q$, we will also require satisfying all three Assumptions 1, 2, 5.

### C.1   Convex Functions

We start with the following lemma, which is a special case of Lemma 6 from [19]. This result can be seen the "optimization" part of the analysis of clipped-SSTM: the proof follows the same steps as the analysis of deterministic Similar Triangles Method [16], [12] and separates stochasticity from the deterministic part of the method.

Pay attention that in full version of Theorem 1 we require Assumptions 1, 2 to hold only on $Q = B_{3R}(x^*)$, however we need to require one more smoothness Assumption 5 as well.

**Theorem 3 (Full version of Theorem 1)** *Let Assumptions 1,2, 5 with $\mu = 0$ hold on $Q = B_{3R}(x^*)$, where $R \geq \|x^0 - x^*\|$, and*

$$a \geq \max \left\{ 48600 \ln^2 \frac{4K}{\beta}, \frac{1800\sigma(K+1)K^{\frac{1}{\alpha}} \ln^{\frac{\alpha-1}{\alpha}} \frac{4K}{\beta}}{B^{\frac{\alpha-1}{\alpha}} LR} \right\}, \tag{27}$$

$$\lambda_k = \frac{R}{30\alpha_{k+1} \ln \frac{4K}{\beta}}, \tag{28}$$

$$L = \frac{M_2 \sqrt{d}}{\tau}, \tag{29}$$

*for some $K > 0$ and $\beta \in (0, 1]$ such that $\ln \frac{4K}{\beta} \geq 1$. Then, after $K$ iterations of* ZO-clipped-SSTM *the iterates with probability at least $1 - \beta$ satisfy*

$$f(y^K) - f(x^*) \leq 2M_2\tau + \frac{6aLR^2}{K(K+3)} \quad and \quad \{x^k\}_{k=0}^{K+1}, \{z^k\}_{k=0}^{K}, \{y^k\}_{k=0}^{K} \subseteq B_{2R}(x^*). \tag{30}$$

*In particular, when parameter $a$ equals the maximum from (27), then the iterates produced by* ZO-clipped-SSTM *after $K$ iterations with probability at least $1 - \beta$ satisfy*

$$f(y^K) - f(x^*) \leq 2M_2\tau + \mathcal{O}\left( \max \left\{ \frac{LR^2 \ln^2 \frac{K}{\beta}}{K^2}, \frac{\sigma R \ln^{\frac{\alpha-1}{\alpha}} \frac{K}{\beta}}{(BK)^{\frac{\alpha-1}{\alpha}}} \right\} \right), \tag{31}$$

*meaning that to achieve $f(y^K) - f(x^*) \leq \varepsilon$ with probability at least $1 - \beta$ with $\tau = \frac{\varepsilon}{4M_2}$* ZO-clipped-SSTM *requires*

$$K = \mathcal{O}\left( \sqrt{\frac{M_2^2\sqrt{d}R^2}{\varepsilon^2}} \ln \frac{M_2^2\sqrt{d}R^2}{\varepsilon^2\beta}, \frac{1}{B} \left(\frac{\sigma R}{\varepsilon}\right)^{\frac{\alpha}{\alpha-1}} \ln \left( \frac{1}{B\beta} \left(\frac{\sigma R}{\varepsilon}\right)^{\frac{\alpha}{\alpha-1}} \right) \right) \quad iterations. \tag{32}$$

*In case when second term in* $\max$ *in (31) is greater total number of oracle calls is*

$$K \cdot B = \mathcal{O}\left( \left(\frac{\sigma R}{\varepsilon}\right)^{\frac{\alpha}{\alpha-1}} \ln \left( \frac{1}{\beta} \left(\frac{\sigma R}{\varepsilon}\right)^{\frac{\alpha}{\alpha-1}} \right) \right).$$

*Proof.* The proof is based on the proof of Theorem F.2 from [35]. We apply first-order algorithm clipped-SSTM for $\frac{M_2\sqrt{d}}{\tau}$-smooth function $\hat{f}_\tau$ with unbiased gradient estimation $g^B$, that has $\alpha$-th moment bounded with $\frac{2\sigma^\alpha}{B^{\alpha-1}}$. Additional randomization caused by smoothing doesn't affect the proof of the original Theorem.

According to it after $K$ iterations we have that with probability at least $1 - \beta$

$$\hat{f}_\tau(y^K) - \hat{f}_\tau(x^*) \leq \frac{6aLR^2}{K(K+3)}$$

and $\{x^k\}_{k=0}^{K+1}, \{z^k\}_{k=0}^{K}, \{y^k\}_{k=0}^{K} \subseteq B_{2R}(x^*)$.

Considering approximation properties of $\hat{f}_\tau$ from Lemma 2

$$f(y^K) - f(x^*) \leq 2M_2\tau + \frac{6aLR^2}{K(K+3)}.$$

Finally, if

$$a = \max \left\{ 48600 \ln^2 \frac{4K}{\beta}, \frac{1800\sigma(K+1)K^{\frac{1}{\alpha}} \ln^{\frac{\alpha-1}{\alpha}} \frac{4K}{\beta}}{B^{\frac{\alpha-1}{\alpha}} LR} \right\},$$

then with probability at least $1 - \beta$

$$
\begin{aligned}
f(y^K) - f(x^*) \;\leq\; & 2M_2\tau + \frac{6aLR^2}{K(K+3)} \\[2mm]
=\; & 2M_2\tau + \max\left\{\frac{291600 LR^2\ln^2\frac{4K}{\beta}}{K(K+3)}, \frac{10800\sigma R(K+1)K^{\frac{1}{\alpha}}\ln^{\frac{\alpha-1}{\alpha}}\frac{4K}{\beta}}{K(K+3)B^{\frac{\alpha-1}{\alpha}}}\right\} \\[2mm]
=\; & 2M_2\tau + \mathcal{O}\left(\max\left\{\frac{LR^2\ln^2\frac{K}{\beta}}{K^2}, \frac{\sigma R\ln^{\frac{\alpha-1}{\alpha}}\frac{K}{\beta}}{(BK)^{\frac{\alpha-1}{\alpha}}}\right\}\right),
\end{aligned}
$$

where $L = \frac{M_2\sqrt{d}}{\tau}$ by Lemma 2.

To get $f(y^K) - f(x^*) \leq \varepsilon$ with probability at least $1 - \beta$ it is sufficient to choose $\tau = \frac{\varepsilon}{4M_2}$ and $K$ such that both terms in the maximum above are $\mathcal{O}(\varepsilon)$. This leads to

$$
K = \mathcal{O}\left(\sqrt{\frac{M_2^2\sqrt{d}R^2}{\varepsilon^2}}\ln\frac{M_2^2\sqrt{d}R^2}{\varepsilon^2\beta}, \frac{1}{B}\left(\frac{\sigma R}{\varepsilon}\right)^{\frac{\alpha}{\alpha-1}}\ln\left(\frac{1}{B\beta}\left(\frac{\sigma R}{\varepsilon}\right)^{\frac{\alpha}{\alpha-1}}\right)\right)
$$

that concludes the proof.

### C.2   Strongly Convex Functions

In the strongly convex case, we consider the restarted version of ZO-clipped-SSTM (R-ZO-clipped-SSTM). The main result is summarized below.

Pay attention that in full version of Theorem 2 we require Assumptions 1, 2 to hold only on $Q = B_{3R}(x^*)$, however we need to require one more smoothness Assumption 5 as well.

**Theorem 4 (Full version of Theorem 2)** *Let Assumptions 1, 2, 5 with $\mu > 0$ hold for $Q = B_{3R}(x^*)$, where $R \geq \|x^0 - x^*\|^2$ and* R-ZO-clipped-SSTM *runs* ZO-clipped-SSTM $N$ *times. Let*

$$
L_t = \frac{\sqrt{d}M_2}{\tau_k}, \quad \tau_k = \frac{\varepsilon_k}{M_2}, \tag{33}
$$

$$
K_t = \left\lceil\max\left\{1080\sqrt{\frac{L_t R_{t-1}^2}{\varepsilon_t}}\ln\frac{2160\sqrt{L_t R_{t-1}^2}N}{\sqrt{\varepsilon_t}\beta}, \frac{2}{B_t}\left(\frac{10800\sigma R_{t-1}}{\varepsilon_t}\right)^{\frac{\alpha}{\alpha-1}}\ln\left(\frac{4N}{B_t\beta}\left(\frac{5400\sigma R_{t-1}}{\varepsilon_t}\right)^{\frac{\alpha}{\alpha-1}}\right)\right\}\right\rceil, \tag{34}
$$

$$
\varepsilon_t = \frac{\mu R_{t-1}^2}{4}, \quad R_{t-1} = \frac{R}{2^{(t-1)/2}}, \quad N = \left\lceil\log_2\frac{\mu R^2}{2\varepsilon}\right\rceil, \quad \ln\frac{4K_t N}{\beta} \geq 1, \tag{35}
$$

$$
a_t = \max\left\{48600\ln^2\frac{4K_t N}{\beta}, \frac{1800\sigma(K_t+1)K_t^{\frac{1}{\alpha}}\ln^{\frac{\alpha-1}{\alpha}}\frac{4K_t N}{\beta}}{B_t^{\frac{\alpha-1}{\alpha}}L_t R_t}\right\}, \tag{36}
$$

$$
\lambda_k^t = \frac{R_t}{30\alpha_{k+1}^t\ln\frac{4K_t N}{\beta}} \tag{37}
$$

*for $t = 1, \ldots, \tau$. Then to guarantee $f(\hat{x}^\tau) - f(x^*) \leq \varepsilon$ with probability $\geq 1 - \beta$* R-clipped-SSTM *requires*

$$
\mathcal{O}\left(\max\left\{\sqrt{\frac{M_2^2\sqrt{d}}{\varepsilon\mu}}\ln\left(\frac{\mu R^2}{\varepsilon}\right)\ln\left(\frac{M_2 d^{\frac{1}{4}}}{\sqrt{\mu\varepsilon}\beta}\ln\left(\frac{\mu R^2}{\varepsilon}\right)\right), \left(\frac{\sigma^2}{\mu\varepsilon}\right)^{\frac{\alpha}{2(\alpha-1)}}\ln\left(\frac{1}{\beta}\left(\frac{\sigma^2}{\mu\varepsilon}\right)^{\frac{\alpha}{2(\alpha-1)}}\ln\left(\frac{\mu R^2}{\varepsilon}\right)\right)\right\}\right) \tag{38}
$$

*oracle calls. Moreover, with probability $\geq 1 - \beta$ the iterates of* R-clipped-SSTM *at stage $t$ stay in the ball $B_{2R_{t-1}}(x^*)$.*

*Proof.* The proof itself repeats of the proof Theorem F.3 from [35]. In this theorem authors prove convergence of restarted clipped-SSTM. In our case it is sufficient to change clipped-SSTM to ZO-clipped-SSTM and put results of Theorem 3 in order to guarantee $\varepsilon$-solution after $\sum\limits_{t=1}^{N} K_t$ successive iterations.

It remains to calculate the overall number of oracle calls during all runs of clipped-SSTM. We have

$$
\sum_{t=1}^{N} B_t K_t =
$$

$$
= \mathcal{O}\left( \sum_{t=1}^{N} \max\left\{ \sqrt{\frac{M_2^2 \sqrt{d} R_{t-1}^2}{\varepsilon_t^2}} \ln\left( \frac{\sqrt{M_2^2 \sqrt{d} R_{t-1}^2} N}{\varepsilon_t \beta} \right), \frac{1}{B}\left( \frac{\sigma R_{t-1}}{\varepsilon_t} \right)^{\frac{\alpha}{\alpha-1}} \ln\left( \frac{N}{B\beta}\left( \frac{\sigma R_{t-1}}{\varepsilon_t} \right)^{\frac{\alpha}{\alpha-1}} \right) \right\} \right)
$$

$$
= \mathcal{O}\left( \sum_{t=1}^{N} \max\left\{ \sqrt{\frac{M_2^2 \sqrt{d}}{R_{t-1}^2 \mu^2}} \ln\left( \frac{\sqrt{M_2^2 \sqrt{d}} N}{\mu R_{t-1} \beta} \right), \left( \frac{\sigma}{\mu R_{t-1}} \right)^{\frac{\alpha}{\alpha-1}} \ln\left( \frac{N}{\beta}\left( \frac{\sigma}{\mu R_{t-1}} \right)^{\frac{\alpha}{\alpha-1}} \right) \right\} \right)
$$

$$
= \mathcal{O}\left( \max\left\{ \sum_{t=1}^{N} 2^{t/2} \sqrt{\frac{M_2^2 \sqrt{d}}{R^2 \mu^2}} \ln\left( 2^{t/2} \frac{\sqrt{M_2^2 \sqrt{d}} N}{\mu R \beta} \right), \sum_{t=1}^{N} \left( \frac{\sigma \cdot 2^{t/2}}{\mu R} \right)^{\frac{\alpha}{\alpha-1}} \ln\left( \frac{N}{\beta}\left( \frac{\sigma \cdot 2^{t/2}}{\mu R} \right)^{\frac{\alpha}{\alpha-1}} \right) \right\} \right)
$$

$$
= \mathcal{O}\left( \max\left\{ \sqrt{\frac{M_2^2 \sqrt{d}}{R^2 \mu^2}} 2^{N/2} \ln\left( 2^{N/2} \frac{\sqrt{M_2^2 \sqrt{d}}}{\mu R \beta} \ln\left( \frac{\mu R^2}{\varepsilon} \right) \right), \left( \frac{\sigma}{\mu R} \right)^{\frac{\alpha}{\alpha-1}} \ln\left( \frac{N}{\beta}\left( \frac{\sigma \cdot 2^{N/2}}{\mu R} \right)^{\frac{\alpha}{\alpha-1}} \right) \sum_{t=1}^{N} 2^{\frac{\alpha t}{2(\alpha-1)}} \right\} \right)
$$

$$
= \mathcal{O}\left( \max\left\{ \sqrt{\frac{M_2^2 \sqrt{d}}{\varepsilon \mu}} \ln\left( \frac{\sqrt{M_2^2 \sqrt{d}}}{\sqrt{\varepsilon \mu} \beta} \ln\left( \frac{\mu R^2}{\varepsilon} \right) \right), \left( \frac{\sigma}{\mu R} \right)^{\frac{\alpha}{\alpha-1}} \ln\left( \frac{N}{\beta}\left( \frac{\sigma}{\mu R} \right)^{\frac{\alpha}{\alpha-1}} \cdot 2^{\frac{\alpha N}{2(\alpha-1)}} \right) 2^{\frac{\alpha N}{2(\alpha-1)}} \right\} \right)
$$

$$
= \mathcal{O}\left( \max\left\{ \sqrt{\frac{M_2^2 \sqrt{d}}{\varepsilon \mu}} \ln\left( \frac{\sqrt{M_2^2 \sqrt{d}}}{\sqrt{\varepsilon \mu} \beta} \ln\left( \frac{\mu R^2}{\varepsilon} \right) \right), \left( \frac{\sigma^2}{\mu \varepsilon} \right)^{\frac{\alpha}{2(\alpha-1)}} \ln\left( \frac{1}{\beta}\left( \frac{\sigma^2}{\mu \varepsilon} \right)^{\frac{\alpha}{2(\alpha-1)}} \ln\left( \frac{\mu R^2}{\varepsilon} \right) \right) \right\} \right),
$$

which concludes the proof.

