# OpenReview forum: "Accelerated Zeroth-order Method for Non-Smooth Stochastic Convex Optimization Problem with Infinite Variance"
_NeurIPS.cc/2023/Conference — NeurIPS 2023 poster_

### Official Review · Reviewer_zmkR · 2023-06-25

**Soundness:** 3 good
**Presentation:** 2 fair
**Contribution:** 2 fair
**Rating:** 6
**Confidence:** 3

**Summary:**

This paper proposes a novel gradient-free (zeroth-order) clipped version of stochastic similar triangles method for solving non-smooth stochastic convex optimization problem under a much weaker infinite variance assumption. The derived iteration and oracle complexity bounds are optimal in both convex and strongly convex case.

**Strengths:**

The problem addressed in this paper is well-motivated, the write-up is easy to follow, and the novelty and contribution are easy to identify.

**Weaknesses:**

1. While the paper is a valuable theoretical contribution, the addition of experimental results would enhance the overall work by demonstrating the feasibility and effectiveness of this method in practical applications.

2. The absence of a Conclusion section or any definitive end to the work makes it feel like a work in progress.

**Questions:**

Line 118: Should the upper bound of the second inequality be a notation $\sigma_B^{\alpha}$?

---

> ### Author Rebuttal · Authors · 2023-08-08
>
> >**While the paper is a valuable theoretical contribution, the addition of experimental results would enhance the overall work by demonstrating the feasibility and effectiveness of this method in practical applications.**
>
> Thank you for the suggestion. Please, see our general response to all reviewers where we provided the results of numerical experiments and also describe them.
>
> >**The absence of a Conclusion section or any definitive end to the work makes it feel like a work in progress.**
>
> Thank you for the suggestion. We will add a conclusion section that will summarize our work, briefly list our contributions and reveal possible directions for further research.
>
>
> >**Line 118: Should the upper bound of the second inequality be a notation $\sigma_B^\alpha$**
>
>
> Thank you for the question. We do not need to introduce $\sigma_B$ since its value is explicitly written in the third formula in Lemma 3. We will remove $\sigma_B$ in the final version.

---

> ### Comment · Reviewer_zmkR · 2023-08-13
>
> I appreciate the authors' response, which adequately resolved my concerns. Due to the addition of the experimental and conclusion chapters, I will raise my score to 6.

---

> > ### Author Response · Authors · 2023-08-14
> > **Thank you for the response**
> >
> > We are glad that the reviewer's concerns were resolved and are grateful for raising the score.

---

### Official Review · Reviewer_M3Ug · 2023-07-05

**Soundness:** 3 good
**Presentation:** 2 fair
**Contribution:** 2 fair
**Rating:** 5
**Confidence:** 3

**Summary:**

In this papers, the authors build upon the work that has been done in [28] and adjust the proposed algorithms there for zero-order oracles rather than the gradient oracles. The goal is to optimize non-smooth stochastic convex optimization problems with infinite variance.

**Strengths:**

The paper does a good job when it comes to introducing the notions they have used with clarity. The organization of the paper helps the reader to understand the concepts. The quality of the write up and the technical contributions look solid.

**Weaknesses:**

- Minor typo in the abstract: ajust --> adjust.

- "We emphasis (--> emphasize) that this generalization requires an extension of the batching technique to (--> for) infinite variance." The emphasis should be emphasize in this sentence and I believe for is more suitable than to.

- The motivation about why we should be interested in such problems is only explained in citation numbers ([30, 6]) with no mention of what these motivating examples actually are. You have 9 pages of space without the references and you are not utilizing all of it. I suggest using the remaining space to attract more attention to your paper by mentioning concrete examples. This would help with the visibility of your paper as well.

- "• can be generalized for saddle-point problems (based on [28]) and one-point feedback [13]. We leave it for future work." This should not been mentioned in the contributions since this paper has not actually made this contributions yet. You can add a conclusion and/or future work paragraph or section in the end mention it there.

- Similarly, related work could have a section of its own.

- My main concern about this paper is that, even though adjusting the method from [28] for two-point zero-order oracle is nontrivial, it feels a bit too incremental for a venue like NeurIPS. Also, the trade-off between using weaker or stronger assumptions can be discussed further in order to convince the reader on the advantages and the disadvantages of the zeroth-order methods and where to use which.

**Questions:**

- In which cases we should utilize the proposed accelaerated zeroth-order method for non-smooth stochastic convex optimization problems with infinite variance?

- It is not very clear to me why the paper considers the case when \delta = 0 on page 6. Could you please elaborate more on that?

- Could you please list some instances where non-smooth stochastic convex optimization problems show up and why we should be interested in them?

**Limitations:**

The advantages and the disadvantages of the proposed method is not discussed in detail except for the advantage of enabling weaker assumptions.

---

> ### Author Rebuttal · Authors · 2023-08-08
>
> We thank the reviewer for a detailed review of our work. Below, we address questions and concerns raised by the reviewer.
>
> >**Minor typo in the abstract: ajust --> adjust.**
>
> >**"We emphasis (--> emphasize) that this generalization requires an extension of the batching technique to (--> for) infinite variance." The emphasis should be emphasize in this sentence and I believe for is more suitable than to.**
>
> Thank you, we will fix these typos.
>
> > **Motivating examples.**
>
> Thank you for the suggestion. See our general response where we provide motivating examples. We will definitely add them to the final version.
>
>
> >**"• can be generalized for saddle-point problems (based on [28]) and one-point feedback [13]. We leave it for future work." This should not been mentioned in the contributions since this paper has not actually made this contributions yet. You can add a conclusion and/or future work paragraph or section in the end mention it there.**
>
> You are right, thank you for the comment. We will move this part to the conclusion section.
>
> >**Similarly, related work could have a section of its own.**
>
> We agree with the reviewer and promise to fix the final version of the paper accordingly.
>
>
> >**My main concern about this paper is that, even though adjusting the method from [28] for two-point zero-order oracle is nontrivial, it feels a bit too incremental for a venue like NeurIPS. Also, the trade-off between using weaker or stronger assumptions can be discussed further in order to convince the reader on the advantages and the disadvantages of the zeroth-order methods and where to use which.**
>
>
> Thank you for this suggestion. We should definitely more thoroughly explain the need for zero-order methods in the introduction. Zero-order methods should be used when one has black-box access to an objective function, e.g., the objective function is computed by a black-box simulation package or it is the result of a real experiment. Thus, automatic differentiation is impossible. This is often the case of various problems encountered in medicine, biologics, physics, and etc. The disadvantage of zero-order methods is the dependence of iteration and oracle complexity on the problem dimension $d$. The main advantage of zero-order methods is the possibility to solve an optimization problem when it is impossible to apply first-order methods (as we cannot calculate derivatives). However, all existing zero-order methods are not robust to heavy-tailed noise, which is why we propose a zero-order algorithm that is able to cope with this issue.
>
> Our main result indeed relies on the techniques from Sadiev et al. (2023), but we would like to highlight that the adjustment of clipped-SSTM to the derivative-free setup is non-trivial. To do it we needed to generalize the smoothing technique to the case of bounded $\alpha$-th moment, e.g., see Lemma 11. Next, to achieve optimal oracle and iteration complexities, we also needed to generalize the classical result about batching from bounded variance to bounded $\alpha$-th moment case. This results in Lemma 9 that is interesting on its own. To the best of our knowledge, the previous result of this type is Lemma 7 from [1] that has an extra factor of $d^{1-\frac{\alpha}{2}}$, where $d$ is the dimension of the problem. For huge-scale problems, this factor can be large even for $\alpha \approx 3/2$. In contrast, our Lemma 9 is dimension-independent.
>
> [1] Wang et al. Convergence rates of stochastic gradient descent under infinite noise variance. NeurIPS 2021.
>
> >**In which cases we should utilize the proposed accelaerated zeroth-order method for non-smooth stochastic convex optimization problems with infinite variance?**
>
>
> We refer to our general comment to the reviewers (see the section with motivating examples).
>
> >**It is not very clear to me why the paper considers the case when \delta = 0 on page 6. Could you please elaborate more on that?**
>
>
> We thank the reviewer for spotting this typo. We will fix it in the final version. We focus mostly on the case of $\Delta = 0$ to make the proofs simpler and more readable. The proofs for the case of $\Delta > 0$ follow similar steps and require just accurate calculations. In particular, the additive noise creates extra non-stochastic bias terms in the sums like (12). Since the noise $\delta(x)$ is bounded, these bias terms can be upper-bounded by induction using the same technique as in the proof for the case $\Delta = 0$ (the idea is described in lines 232-244). We will add these details to the final version.
>
> >**Could you please list some instances where non-smooth stochastic convex optimization problems show up and why we should be interested in them?**
>
>
> Stochastic non-smooth convex optimization problems arise in machine learning (population risk minimization) and statistical application (e.g., likelihood maximization).  We can list SVM or ReLU activation functions in deep learning problems. See also our general response for motivating examples.

---

> > ### Comment · Reviewer_M3Ug · 2023-08-17
> >
> > I thank the authors for carefully responding to my concerns.

---

> > > ### Author Response · Authors · 2023-08-19
> > > **Thank you for the response**
> > >
> > > We thank the reviewer for checking our response and for the positive rating.

---

### Official Review · Reviewer_FKLk · 2023-07-07

**Soundness:** 4 excellent
**Presentation:** 2 fair
**Contribution:** 3 good
**Rating:** 7
**Confidence:** 4

**Summary:**

This paper proposed and analyzed a zeroth-order method for non-smooth stochastic optimization under heavy-tailed noise and adversarial noise, by combining ball-averaging-based smoothing technique (to tackle non-smoothness) and gradient clipping technique (to tackle heavy-tailed/adversarial noise). This generalizes previous results where $L_\infty$ or $L_2$ boundness of noise is assumed. There are also several technical improvements over previous results, including proving a high-probability bound instead of a bound in expectation.

**Strengths:**

1. The topic is important. Gradient free method for stochastic optimization is a popular field of research, and it certainly helps to handle the heavy-tailed/adversarial noisy setting which is common in practice.
2. I have briefly gone through the proof and have no doubt on its correctness.
3. Though not proved in the paper, the theoretical bound seems close to being tight.

**Weaknesses:**

1. The presentation is concise, but maybe at the cost of some necessary clarity. In Eqn. (2) it's not specified what distribution should $\bf e$ follow (should be the uniform distribution in the unit _ball_), which leaves the entire equation undefined. It becomes even more confusing that later in (3) $\bf e$ is used with a different meaning, denoting a random vector uniformly distributed on the unit _sphere_!
2. It may also improve clarity to discuss the order of the theoretical bounds and compare them with previous results. In particular, it would be helpful to discuss whether the dependence of $\alpha$ is optimal.

**Questions:**

I have no additional questions other than the ones discussed in `Weaknesses`.

**Limitations:**

Yes.

---

> ### Author Rebuttal · Authors · 2023-08-08
>
> We thank the reviewer for a positive evaluation of our work. Below, we address questions and concerns raised by the reviewer.
>
> >**The presentation is concise, but maybe at the cost of some necessary clarity. In Eqn. (2) it's not specified what distribution should $e$ follow (should be the uniform distribution in the unit ball), which leaves the entire equation undefined. It becomes even more confusing that later in (3) $e$ is used with a different meaning, denoting a random vector uniformly distributed on the unit sphere!**
>
> We thank the reviewer for spotting this mismatch in our notation. Eqn. (2) is the only place where vector $e$ should be uniformly distributed on the unit **ball**. Everywhere else $e$ is a random vector uniformly distributed on the unit **sphere**. To avoid this confusion, we propose the following change that we will apply in the final version: in Eqn. (2) we will use vector $u$ to denote a random vector uniformly distributed on the unit ball. Then, everywhere in the paper, vector $e$ will denote a random vector uniformly distributed on the unit sphere.
>
> >**It may also improve clarity to discuss the order of the theoretical bounds and compare them with previous results. In particular, it would be helpful to discuss whether the dependence of $\alpha$ is optimal.**
>
> Our work is the first work on gradient-free optimization with heavy-tailed noise.  Optimality of oracle complexity in terms of the dependence on $d$ is an open problem in non-smooth settings (if $\alpha$=2, i.e., when noise has bounded variance, it is optimal). However, it is not optimal for **smooth** stochastic convex optimization problems with $(d+1)$-points stochastic zero-order oracle.
> Iteration complexity and maximal level of noise are optimal (coincides with lower bounds in one of the regimes, see [1, 2]). We will add these remarks to the final version of the paper.
>
> [1]  Bubeck, S., & Cesa-Bianchi, N. (2012). Regret analysis of stochastic and nonstochastic multi-armed bandit problems. Foundations and Trends® in Machine Learning, 5(1), 1-122.
>
> [2]   A. Risteski and Y. Li. Algorithms and matching lower bounds for approximately-convex optimization. Advances in Neural Information Processing Systems, 29:4745–4753, 2016.

---

> > ### Comment · Reviewer_FKLk · 2023-08-12
> >
> > I appreciate the authors' response and will keep my score unchanged.

---

> > > ### Author Response · Authors · 2023-08-14
> > > **Thank you for the response**
> > >
> > > We thank the reviewer for checking our response and for the very positive evaluation.

---

### Official Review · Reviewer_nSeP · 2023-07-07

**Soundness:** 3 good
**Presentation:** 4 excellent
**Contribution:** 3 good
**Rating:** 7
**Confidence:** 3

**Summary:**

This paper presented derivative-free methods for the optimization of stochastic convex functions with a potentially infinite variance noise. Here the level of noise is defined in terms of the boundedness of modulus of a Hölder-type continuous condition. The main technique is to adopt a gradient clipping to the two-point estimation of the gradient of the randomized smoothed function. For some of their results, they also claim the attained bounds are rate optimal. The presentation and organization is of very clear and relatively easy to follow.

**Strengths:**

Overall the contribution is well-motivated, and fits into the flurry of recent development of methods for problem with infinite noise variance. The paper is well written, and the proofs are intuitive and relatively easy to follow.


**Weaknesses:**

I only have some minor comments and questions:

* L113: it might be better to use {\xi_i}_i and {e_i}_i as the input of the function g^B.
* L119: where is the \sigma_B used in the statement of Lemma 3?
* the second line of Section 2: you might need to assume g \neq 0.
* L126: where is the distribution D_k defined?
* L139: could you elaborate on why the first term is optimal using the lower bound from [4]? Do you need some assumption on batch-size B here to illustrate rate optimality?
* L152: it might be better to emphasize that w.p. 1-\beta, *for any* 1 \leq t \leq N, ..., if the boundedness of iteration holds uniformly.
* Question (out of curiosity): is the bound in L190 optimal? Or, is there any lower bound for such a so-called maximum allowable noise level?
* L224: proof -> prove.

**Questions:**

See above.

**Limitations:**

Yes.

---

> ### Author Rebuttal · Authors · 2023-08-08
>
> We thank the reviewer for a positive evaluation of our work. Below, we address questions and concerns raised by the reviewer.
>
>
>
> >**L113: it might be better to use {\xi_i}_i and {e_i}_i as the input of the function g^B.**
>
>
> Thank you for the suggestion, we agree with it. We did not want to complicate formulas with $g^B$ and thus used this notation in the paper. We can use bold symbols with superscript $B$ so that there was no confusion: $\mathbf{e}^B = \lbrace e_i\rbrace_i$ and $\boldsymbol{\xi}^B =\lbrace\xi_i\rbrace_i$. Then we were able to use $\mathbf{e}^B$ and as $\boldsymbol{\xi}^B$ the input of the function $g^B(x, \mathbf{e}^B, \boldsymbol{\xi}^B)$. We will change this in the final version of our paper.
>
>
> >**L119: where is the \sigma_B used in the statement of Lemma 3?**
>
>
> Thank you for the question. We do not need to introduce $\sigma_B$ since its value is explicitly written in the third formula in Lemma 3. We will remove $\sigma_B$ in the final version.
>
> >**the second line of Section 2: you might need to assume g \neq 0.**
>
>
> Thank you, we will add this and also add that $\text{clip}(0,\lambda) = 0$.
>
>
> >**L126: where is the distribution D_k defined?**
>
>
> Thank you, this is a typo. It must be $D$. This is a distribution of random variable $\xi$. We do not know $D$ but we can sample from it (see Eq. (1) in line 14).
>
>
> >**L139: could you elaborate on why the first term is optimal using the lower bound from [4]? Do you need some assumption on batch-size B here to illustrate rate optimality?**
> This optimality is in terms of $\varepsilon$. The first term is independent of batch size B. In **[4]** (citation is from our work), it was proven that this corresponds to the lower bound in one of the regimes in a noiseless setup. The additional presence of noise cannot improve iteration and oracle complexity, it can only make it worse (due to its possible adversarial nature). The number of iterations of our method that allows the presence of (possibly non-stochastic) noise corresponds to the lower bound for a noiseless setup, that is, the number of iterations is optimal. Also, we refer to [1] where the authors propose an optimal algorithm in terms of oracle and iteration complexity but in classical settings of noise with bounded variance. The iteration complexity of the method from [1] coincides with the iteration complexity of our method. Thus, our algorithm can be also seen as a robust version of the algorithm from [1] that makes it possible to work with heavy-tailed noise.
> **[4]** Sébastien Bubeck, Qijia Jiang, Yin-Tat Lee, Yuanzhi Li, and Aaron Sidford. Complexity of highly parallel non-smooth convex optimization. Advances in neural information processing systems, 32, 2019.
>
> [1] Gasnikov, A., Novitskii, A., Novitskii, V., Abdukhakimov, F., Kamzolov, D., Beznosikov, A., ... & Gu, B. (2022, June). The power of first-order smooth optimization for black-box non-smooth problems. In International Conference on Machine Learning (pp. 7241-7265). PMLR.
>
>
> >**L152: it might be better to emphasize that w.p. 1-\beta, for any 1 \leq t \leq N, ..., if the boundedness of iteration holds uniformly.**
>
>
> Thank you, we will do it.
>
>
> >**Question (out of curiosity): is the bound in L190 optimal? Or, is there any lower bound for such a so-called maximum allowable noise level?**
>
> The lower bound for $\Delta$ (maximal absolute value of noise) is given in [1]. Our bound exactly corresponds to that lower bound in the regime when $\varepsilon^{-2} \lesssim d $. That is the large-dimension regime, when the subgradient method is better than the center of gravity type methods [2].
>
> [1]   A. Risteski and Y. Li. Algorithms and matching lower bounds for approximately-convex optimization. Advances in Neural Information Processing Systems, 29:4745–4753, 2016.
>
> [2] A. S. Nemirovsky and D. B. Yudin. Problem complexity and method efficiency in optimization. Wiley-Interscience, 1983.
>
> >**L224: proof -> prove.**
>
>
> Thank you, we will fix this typo.

---

> > ### Comment · Reviewer_nSeP · 2023-08-18
> >
> > I appreciate the thorough responses, and I'll maintain my current score.

---

> > > ### Author Response · Authors · 2023-08-19
> > > **Thank you for the response**
> > >
> > > We thank the reviewer for checking our response and for the very positive evaluation.

---

### Official Review · Reviewer_6zia · 2023-07-09

**Soundness:** 3 good
**Presentation:** 1 poor
**Contribution:** 3 good
**Rating:** 6
**Confidence:** 3

**Summary:**

This paper provides high probability bounds for the convergence of gradient-free methods on convex and strongly-convex functions when the noise in the gradient oracle has infinite variance. An oracle provides $f(x,\xi)$, a noisy evaluation of the function $f$ at point $x\in \mathbb{R}^d$ by the oracle, where $\xi$ is the noise variable. For the same noise $\xi$, the function is $M_2(\xi)$-Lipschitz, where $M_2(\xi)$ quantifies the noise level.

If the noise variance is finite, ($\mathbb{E}[M_2(\xi)^2] < \infty$), [1] provides optimal iteration and oracle complexity for convergence of in expectation. The primary technique used in [1] is a batched accelerated gradient method which uses smoothing. For a fixed constant $\tau>0$ which defines the smoothing level, smoothing computes an approximate gradient of $f$ at point $x\in \mathbb{R}^d$ as,

$$
g^B(x) = \frac{d}{2B \tau }\sum_{i=1}^B (f(x + \tau e_i, \xi_i) - f(x - \tau e_i, \xi_i))e_i
$$
Here, $e_i$ are sampled uniformly from the unit sphere, $\xi_i$ are the noise variables of the oracle and $B$ is the batch size. On expectation, the smoothed gradient is the gradient of $ \mathbb{E}_{e,\xi}[f(x + \tau e, \xi)] $. For small value of $\tau$, this approximation is close to $f$. Further, even if the function $f$ is non-smooth but Lipschitz, smoothing makes  $\frac{\sqrt{d}M_2}{\tau}$ smooth.



This paper extends this technique to the infinite noise variance setting, $\mathbb{E}[M_2(\xi)^\alpha] < M_2^\alpha$ for some $\alpha \in (1,2]$ by applying clipping. Specifically, the technique of clipped Stochastic Similar Triangles used for handling heavy tailed noise in smooth optimization is extended to the non-smooth case by the above smoothing procedure.

For convex Lipschitz functions, the iteration complexity and oracle complexity is $\frac{\sqrt{d}^{1/4}}{\epsilon}$ and $\left(\frac{\sqrt{d}}{\epsilon}\right)^\frac{\alpha}{\alpha-1}$. For $\mu$-strongly convex and lipshcitz functions, the corresponding bounds are $\frac{d^{1/4}}{(\mu\epsilon)^{-1/2}}$ and $\left(\frac{d}{\mu\epsilon}\right)^{\frac{\alpha}{\alpha-1}}$. These rates are shown to be optimal in $\epsilon$.


Further, when if the oracle provides a corrupted value of $f$ with an additive corruption of $\lvert\delta(x)\rvert \leq \Delta$, the authors derive the maximum possible values of $\Delta$ such that the convergence rates for both smooth and non-smooth settings are unaffected by the corruption.


**References**
1. Gasnikov et al. The power of first-order smooth optimization for black-box non-smooth problems. ICML 2022.
2. Sadiev et al. High-Probability Bounds for Stochastic Optimization and Variational Inequalities: the Case of Unbounded Variance. ICML 2023.

**Strengths:**

- **Interesting Problem Setting**:  Heavy-tailed noise is a significant problem which violates the commonly used bounded variance assumption in stochastic optimization.  The authors extend the solution of clipping to handle it for derivative-free methods.
- **Convergence Rates** : These are the first convergence rates for derivate-free optimization under heavy-tailed noise. Further, the rates are high probability bounds instead of in expectation. Additionally, for both cases, convex and strongly convex,  the obtained rates are optimal in terms of error $\epsilon$.
- **Thorough literature review** : The authors thoroughly review existing results in derivative-free methods and clipping.

**Weaknesses:**

- **Presentation** : The paper seems to be missing an introduction and experiment section. Although the paper is theoretical, the proposed algorithm, clipped-SSTM with two-point feedback, is new and should have been tested on at least synthetic problems.

- **Lack of a motivating example** : The authors do not provide a motivating example which justifies the heavy-tailed noise in gradient-free settings.

**Questions:**

- Are there any lower bounds for oracle and iteration complexity in the adversarial corruption case in terms of $\Delta$?
- Are there methods other than clipping to handle unbounded variance in stochastic optimization?

**Limitations:**

-

---

> ### Author Rebuttal · Authors · 2023-08-08
>
> We thank the reviewer for a detailed summary of our main contributions. Below, we address questions and concerns raised by the reviewer.
>
> > **Missing introduction and numerical experiments.**
>
> We agree that the introductory part can be improved and will extend it in the final version of our work. We will add more motivational examples about the significance of the problem we are solving, in particular, motivational examples that justifies the use of gradient-free methods as well as the need to consider heavy-tailed noise. Moreover, we understand your concern about the presentation of our work, but we promise that the final version will be much more comprehensive and precise.
>
> The numerical experiments are provided in the general response to all reviewers.
>
> > **Motivating examples.**
>
> Thank you for the suggestion. See our general response where we provide motivating examples. We will definitely add them to the final version.
>
> >**On the lower bounds.**
>
> The lower bound for the iteration complexity can be found in [1]. The lower bound for oracle complexity is presented in [1] and [2]. These lower bounds were obtained in a noiseless setup ($\Delta = 0$). The lower bound for $\Delta$ (maximal absolute value of noise) is given in [3]. The additional presence of noise cannot improve iteration and oracle complexity, it can only make it worse. The upper bounds of our method meet (up to the numerical and logarithmic factors) all of these three lower bounds (for iteration complexity and $\Delta$, this holds in the regime when $\varepsilon^{-2} \lesssim d$, this is the large-dimension regime, when the subgradient method is better than the center of gravity type methods [1]). Thus, our algorithm is optimal in terms of all of these three criteria.
>
> [1] Arkadii S. Nemirovsky and David B. Yudin. Problem complexity and method efficiency in optimization. Wiley-Interscience, 1983.
>
> [2]  S. Bubeck, Q. Jiang, Y. T. Lee, Y. Li, A. Sidford, et al. Complexity of
> highly parallel non-smooth convex optimization. Advances in neural information processing systems, 2019.
>
> [3]   A. Risteski and Y. Li. Algorithms and matching lower bounds for approximately-convex optimization. Advances in Neural Information Processing Systems, 29:4745–4753, 2016.
>
>
> >**Alternatives to clipping.**
>
> Heavy-tailed noise can also be handled without explicit gradient clipping. For example, one can use Stochastic Mirror Descent algorithm with a particular class of uniformly convex mirror maps [1]. This algorithm does not require any explicit gradient clipping or normalization. However, the convergence guarantee was given in expectation. Moreover, it is not clear how to apply batching and acceleration for this method. Without this, we would not be able to get the optimal method in terms of the number of iterations and not only in terms of oracle complexity. There are also some studies on the alternatives to gradient clipping [2] but the results for these alternatives are given in-expectation and are weaker than the state-of-the-art results for the methods with clipping. This is another reason why we choose gradient clipping to handle the heavy-tailed noise.
>
> [1] Vural, Nuri Mert, et al. "Mirror descent strikes again: Optimal stochastic convex optimization under infinite noise variance." Conference on Learning Theory. PMLR, 2022.
>
> [2] Jakovetić, Dus̆an, et al. "Nonlinear gradient mappings and stochastic optimization: A general framework with applications to heavy-tail noise." SIAM Journal on Optimization 33.2 (2023): 394-423.

---

> > ### Comment · Reviewer_6zia · 2023-08-18
> > **Response**
> >
> > Thanks for providing a detailed response to all of my questions. The motivating example and numerical experiments seem nice and the authors should include it in the final version of the draft. I'm increasing my score based on this.

---

> > > ### Author Response · Authors · 2023-08-19
> > > **Thank you for the response**
> > >
> > > We are very grateful to the reviewer for raising the score. We will definitely include motivating examples and numerical experiments in the final version.

---

### Author Rebuttal · Authors · 2023-08-08

We thank the reviewers for their valuable feedback and time. In particular, we appreciate that the reviewers acknowledged the following strengths of our work: a well-motivated problem, an algorithm with tight and valid theoretical bounds, an important contribution, and a good write-up and organization.

The reviewers also have several questions and concerns that we address in our responses to each reviewer.

In this general comment, we respond to the common reviewers’ comments and concerns.

## Motivation examples

In machine learning, the interest in gradient-free methods is mainly driven by the bandit optimization problem [1,2]. The vast majority of authors assume sub-Gaussian distribution of rewards. However, in some practical cases (e.g., in finance [3]) rewards distribution has heavy tails or can be adversarial. For the heavy-tailed bandit optimization, we refer to [4].
Moreover, in many applications of medicine, biologics, physics, etc., the objective function is only computable through numerical simulation or the result of a real experiment, i.e., automatic differentiation cannot be employed to calculate function derivatives.  Usually, a black-box function we are optimizing is affected by stochastic or computational noise. This noise can arise naturally from modeling randomness within a simulation or by computer discretization. The classical setting assumes this noise to have light tails. However, usually, in black-box optimization, we know nothing about the function, only its values at requested points are available/computable, so any assumptions about noise may not be fulfilled. And if so, gradient-free algorithms may diverge. We aim to construct an algorithm that is robust even to heavy-tailed noise that does not have finite variance. In theory, one can consider heavy-tailed noise to model a situation when noticeable outliers may occur in practice (even if the nature of these outliers is non-stochastic). That is why we relax classical assumptions about finite variance and consider less burdensome assumptions of finite $\alpha$-th moment.

## Numerical experiments

Following the reviewers’ requests, we conducted numerical experiments with the proposed method. We consider the following convex non-smooth problem: $\min_{x\in \mathbb{R}^{16}}f(x)$, where $f(x) = \frac{1}{500}\|\| Ax - b \|\|_1$ for some matrix $A \in \mathbb{R}^{500 \times 16}$ and vector $b \in \mathbb{R}^{500}$. The stochastic noise is introduced as follows: $f(x,\xi) = f(x) + \langle \xi, x \rangle$, where $\xi$ is a random vector having independent components sampled from the symmetric Levy $\alpha$-stable distribution with $\alpha = 3/2$. This problem satisfies Assumption 1 with $\mu = 0$ and Assumption 2 with $\alpha = 3/2$ and $M_2(\xi) = \|\|A\|\|_1 + \|\| \xi \|\|_2$. We notice that $\mathbb{E}[M_2(\xi)^\alpha]$ is bounded while $\mathbb{E}[M_2(\xi)^2] = +\infty$ due to the choice of $\xi$.

The existing SOTA zeroth-order methods do not use clipping. They are also obtained from the first-order methods using the smoothing technique and two-point feedback oracle. We call the methods obtained this way from SGD and SSTM as ZO-SGD and ZO-SSTM, respectively, and compare them with our method (ZO-clipped-SSTM). The results of the experiments are provided in the PDF attached to the response. As expected, the methods with clipping fail to converge due to the heavy tails in the distribution of the noise, while ZO-clipped-SSTM converges even in such a setup. This numerical experiment illustrates the necessity of clipping (or other non-linear transformations) in zeroth-order methods for handling heavy-tailed noise.

If the reviewers have further questions/concerns/comments, we will be happy to participate in the discussion.

---
References:

[1]  Flaxman, A. D., Kalai, A. T., & McMahan, H. B. (2004). Online convex optimization in the bandit setting: gradient descent without a gradient. arXiv preprint cs/0408007.

[2]  Bubeck, S., & Cesa-Bianchi, N. (2012). Regret analysis of stochastic and nonstochastic multi-armed bandit problems. Foundations and Trends® in Machine Learning, 5(1), 1-122.

[3]  S.T Rachev: Handbook of Heavy Tailed Distributions in Finance: Handbooks in Finance, Book 1. Elsevier, North Holland (2003)

[4] Dorn, Y., Nikita, K., Kutuzov, N., Nazin, A., Gorbunov, E., & Gasnikov, A. (2023). Implicitly normalized forecaster with clipping for linear and non-linear heavy-tailed multi-armed bandits. arXiv preprint arXiv:2305.06743.

---

### Decision · Program_Chairs · 2023-09-21

**Decision:**

Accept (poster)

**Comment:**

Overview: This paper presented derivative-free methods for the optimization of stochastic convex functions with a potentially infinite variance noise. Here the level of noise is defined in terms of the boundedness of modulus of a Hölder-type continuous condition. The main technique is to adopt a gradient clipping to the two-point estimation of the gradient of the randomized smoothed function. For some of their results, they also claim the attained bounds are rate optimal. The presentation and organization is of very clear and relatively easy to follow.

Pros:

- Instead of listing, the positive points are mostly within all the reviews included in the reviewing process. Interesting Problem Setting, theory of
Convergence Rates,  Thorough literature review

Cons:

- Only disadvantage the lack of a motivating example, that was handled during the discussion.

Overall: Based on the initial reviews + discussion with the authors, the paper has only positive comments, and all the comments raised by the reviewers were adequately tackled by the authors.

The only requirement for the authors is to include (if possible and if space allows) any additional discussion that is useful during the rebuttal phase, in order to improve the paper. Please do not include material that is not presented during the rebuttal and material that cannot be checked, unless another round of review is required.